# SCALING OVERHEAD MATTERS: SALIENCY-AWARE GRAPH-BASED EFFICIENT POST-TRAINING QUANTIZATION FOR LLMS

## ABSTRACT

Post-training quantization (PTQ) is essential for efficiently deploying large language models (LLMs) in resource-constrained environments. Recent PTQ methods achieved near-binary precision. However, existing binarization methods rely on position-based heuristics or fixed saliency assumptions, leading to untracked scaling overhead and limited adaptability across model architectures. We propose SAGE-PTQ (Saliency-Aware Graph-based Efficient PTQ), a novel approach for arbitrary-bit quantization of LLMs. Our formulation comprises five key components: (1) Saliency-Aware Weight Filtering: identifies salient weights based on weight distribution statistics; (2) Affinity-Based Weight Grouping: Models inlier weights with a subsampled graph structure to capture attention patterns and determine the optimal number of weight groups; (3) Dual-Mode Quantizer Optimization: iteratively optimizes weight matrix quantization, minimizing scaling overhead by assigning a single per-channel scale to multi-bit salient weights and a scalar per-group scale to binarized inlier weights; (4) Adaptive Saliency Thresholding: dynamically adjusts the saliency percentage to optimally minimize quantization error; (5) Efficient Inference Runtime: implements a layer-wise lookup to efficiently load binarized weights for accelerated inference. Our approach achieves an average of 1.03 weight bits and 0.004 scaling bits per matrix, significantly outperforming State-of-the-Art schemes like BiLLM and PB-LLM binarization. Evaluations on LLaMA-2-7B yield perplexity of 5.87 (vs. 32.48 for BiLLM) on WikiText2. Compared to BiLLM, our method uses less than 50% of device memory and only 6.5% of the FP16 model size, enabling 1.5× faster token decoding on LLaMA-2-70B with a single NVIDIA L40 GPU. This demonstrates strong potential for efficient inference on edge devices.

## 1 INTRODUCTION

Large language models (LLMs) such as OPT Zhang et al. (2022), LLaMA Touvron et al. (2023a;b), and DeepSeek AI (2024) have shown strong performance in language and reasoning tasks Chen et al. (2021); Wei et al. (2022a); OpenAI (2023). However, LLM size imposes heavy memory demands (e.g., LLaMA-2-70B requires over 140 GB in FP16), hindering edge deployment. Inference further relies on memory-bound GEMV operations during auto-regressive decoding Zeng et al. (2025), which incur repeated DRAM–GPU transfers and increase latency.

To address memory and latency bottlenecks in LLMs, compression techniques such as quantization Liu et al. (2022); Jiao et al. (2019), pruning Frantar et al. (2023); Ma & et al. (2023), and distillation Gou et al. (2021); Tunstall et al. (2023) have been explored. Quantization is favored for efficient deployment of LLMs without retraining. Early PTQ methods achieved good fidelity at 8- and 4-bit precision Dettmers et al. (2022); Xiao et al. (2023); Frantar et al. (2022), and more recent approaches have pushed into extremely low-bit regimes. GPTQ Frantar et al. (2022) extends PTQ to 3 bits but degrades at lower precision. Hybrid binarized PTQ methods, such as PB-LLM Shang et al. (2023) and BiLLM Huang et al. (2024), preserve salient (outlier) weights in higher precision while binarizing inliers, but rely on rigid grouping and heuristic thresholds. These assumptions often fail to generalize across models. We study these limitations by analyzing weight distribution statistics and outlier patterns across diverse LLM families (see appendix), and generalize to four core

issues in PTQ: *(1) Saliency handling*:Outliers vary across layers and require adaptive high-precision treatment. GPTQ Frantar et al. (2022) poorly handles outliers due to calibration dependence,while PB-LLM Shang et al. (2023) applies fixed thresholds across layers.*(2) Weight grouping*:Rigid or uniform grouping fails on long-tailed distributions.BiLLM Huang et al. (2024) assumes Gaussian weights, limiting generalization across architectures and datasets. *Quantization schemes (3)*: Most methods rely on simple min–max scaling, leading to large quantization error. Sub-2-bit methods such as GPTQ, PB-LLM, and BiLLM further assume block-based scaling, introducing untracked memory overheads that can exceed 1 bit per weight and offset the benefits of binarization. *Lookup efficiency*:Efficient group index restoration is critical for deployment, yet BiLLM Huang et al. (2024) and PB-LLM Shang et al. (2023) ignore runtime-aware decoding and lookup, limiting deployment.

**Contributions:** We target the core limitations of Binarized PTQ methods by proposing **SAGE-PTQ**, a saliency-aware, graph-based approach for efficient quantization of LLMs. Our approach consists of the following five modules:

- **We address Saliency handling limitation using Module 1**: *saliency-guided filtering* **and Module 4**: *adaptive saliency thresholding*. Module 1 performs element-wise, statistically informed outlier detection per weight matrix, to estimate an upper bound of salient weights requiring higher precision. Module 4 then refines this through numerical optimization, selecting the exact number of outliers to minimize quantization error.
- **We solve Weight grouping problem using Module 2**: *Graph-based weight grouping* . Uses a sparse KNN graph over subsampled inlier weights to capture attention patterns and determine the per-matrix group count, which is then used for distribution-based clustering and scaling.
- **We address suboptimal Quantization schemes through Module 3**: *dual-mode quantizer design*. This module assigns arbitrary bit precision to salient weights and binarizes inliers, minimizing scaling overhead using a single scalar per inlier group and one channel-wise scale per salient group. For a 4-bit outlier budget, our quantizer achieves 1.03 average bits and only 0.004 scaling bits, outperforming BiLLM with over $5\times$ better perplexity on LLaMA models.
- **We improve deployment practicality using Module 5**: *efficient inference runtime*, separates lookup metadata and enables bit-packed, layer-wise decoding, reducing lookup costs by 25% compared to BiLLM and achieving a 1.5× speedup on 70B models deployed on a single GPU.

The modular structure of SAGE-PTQ method is shown in Figure 1. The SAGE-PTQ method is model-agnostic and hardware-aware, ensuring generalizability. We have validated method performance across LLaMA, OPT, Vicuna, and DeepSeek with sizes: (1.3B–70B), consistently achieve state-of-the-art (SoTA) results with practical deployability.

## 2 RELATED WORK

### 2.1 LLM POST-TRAINING QUANTIZATION

Post-training quantization (PTQ) enables efficient LLM deployment by compressing models into low-bit formats without full retraining. Early PTQ methods targeted weight-only quantization using rounding optimization or compensation Nagel et al. (2020); Li et al. (2021); Frantar et al. (2022), while later approaches extended to activations via scaling and incoherence modeling Yao et al. (2022); Xiao et al. (2023); Tseng et al. (2024). Recent efforts introduced structural priors and learned quantization Kalyvianaki et al. (2011); Kim et al. (2023); Shao et al. (2023) and block-clustered approaches Elangovan et al. (2025), yet binarization remains challenging. Hybrid methods Shang et al. (2023); Huang et al. (2024) retain salient weights in high precision but rely on heuristics and incur overhead. Our method, SAGE-PTQ, addresses these limitations by jointly optimizing weight grouping and precision allocation, achieving SoTA binarized PTQ results across diverse LLM families

### 2.2 SALIENCY-AWARE PRECISION ALLOCATION

Outlier-aware quantization improves model fidelity by assigning higher precision to salient weights identified via metrics like magnitude, Hessian sensitivity, or reconstruction error. HAWQ Dong et al. (2019) uses Hessian trace for layer-wise bitwidths, while QDrop Wei et al. (2022b) applies stochastic dropout during calibration. For LLMs, AWQ Lin et al. (2024) retains high-activation channels in

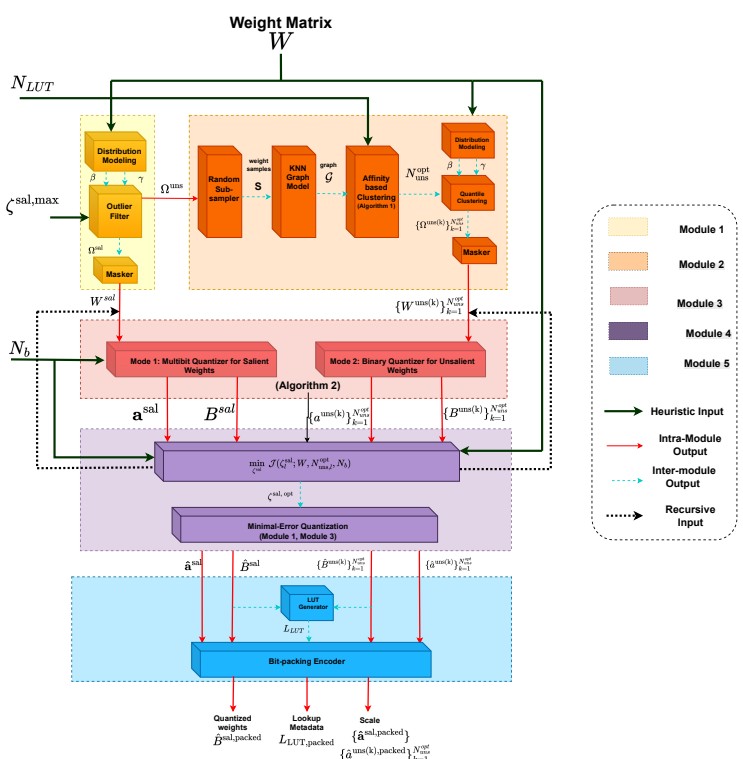

Figure 1: Modular structure of the SAGE-PTQ quantization pipeline.

FP16; SmoothQuant Xiao et al. (2023) rescales outliers; PB-LLM Shang et al. (2023) binarizes inliers and preserves high-magnitude weights in INT8; BiLLM Huang et al. (2024) searches salient columns using residual error. SAGE-PTQ advances this line by combining element-wise saliency filtering with adaptive thresholding to optimally allocate bits to outliers under memory constraints.

## 2.3 GRAPH-BASED CLUSTERING

Graph-based methods such as KNN graphs model local affinities and are widely used in high-dimensional data analysis Zelnik-Manor & Perona (2005); Ng et al. (2002). Spectral clustering von Luxburg (2007) leverages the graph Laplacian to identify coherent groups, with the Silhouette Score Rousseeuw (1987) selecting the optimal cluster count. Prior quantization methods, like vector quantization Gong et al. (2014) and CLIP-Q Tung & Mori (2018), applied clustering without graph structures. SAGE-PTQ introduces graph-based modeling for inlier weights via sparse KNN, applying spectral clustering and Silhouette Score to enable accurate, data-driven quantization groupings.

## 3 THE SAGE-PTQ METHOD

**Preliminaries** We employ a generalized formulation that supports both binarized and multi-bit precision across channels and weight groups. We aproximate a given pretrained weight matrix $W \in \mathbb{R}^{m \times n}$ with $\hat{W}$ by minimizing the reconstruction error:

$$\min_{\hat{W}} \|W - \hat{W}\|_F^2, \quad \text{where} \quad \hat{W} = AB. \tag{1}$$

Here, $A \in \mathbb{R}^{m \times m}$ is a diagonal matrix; can represent channel-level or group-level scaling. The matrix $B \in \mathbb{R}^{m \times n}$ contains $2^{N_b}$ quantized entries where $b_{ij}$ corresponds to a discretized level in $[-1, 1]$, stored efficiently using $N_b$-bit binary encoding for bit-packing. For $N_b = 1$, $b_{ij} \in \{-1, 1\}$, corresponding to standard binarization. Equation equation 1 is based on the Frobenius norm $\| \cdot \|_F$, minimizing the residual energy between $W$ and $\hat{W}$. This yields a compact approximation that is particularly effective when outliers form only a small fraction of the weight.

## 3.1 MODULE 1: SALIENCY-GUIDED FILTERING

Salient weights are highly sensitive to quantization and require higher precision to preserve accuracy. Saliency selection can follow magnitude- or Hessian-based metrics; we study both. Consistent with Shang et al. (2023), our initial analysis shows that magnitude-based saliency more reliably identifies outliers (see appendix), minimizing quantization error and scaling distortion. We analyze weight distributions across diverse model layers. Most layers exhibit Gaussian-like trends under the law of large numbers (LLN), while early layers deviate due to heavy-tailed outliers. Generalizing the approach of Wang et al. (2025), we adaptively select a saliency threshold $\zeta_l^{\text{sal}}$ for each weight matrix $l \in \{1, \ldots, N_l\}$, capped by a heuristic upper bound $\zeta_l^{\text{sal,max}}$. This bound is set by outlier storage budget. The choice of $\zeta_l^{\text{sal}}$ has been studied in Wang et al. (2025). This partitions weight value set $\Omega_l$ into salient and non-salient subsets: $\Omega_l = \Omega_l^{\text{sal}} \cup \Omega_l^{\text{uns}}, \quad \Omega_l^{\text{sal}} \cap \Omega_l^{\text{uns}} = \varnothing$. The salient set is $\Omega_l^{\text{sal}} = \{w \in \Omega_l \mid |w| > \beta_l + \gamma_l \, \Phi^{-1}(\zeta_l^{\text{sal}}/2)\}$, where $\beta_l$ and $\gamma_l$ are the mean and standard deviation of $\Omega_l$, and $\Phi^{-1}(\cdot)$ is the probit function; empirical analysis shows weights are symmetric around $\beta_l$ (see appendix).

## 3.2 MODULE 2: SPARSE GRAPH MODELING FOR AFFINITY-BASED WEIGHT GROUPING

Unsalient weights, targeted for aggressive quantization, often exhibit non-uniform magnitude distributions that challenge uniform min–max schemes. Channel-wise quantization per unsalient group introduces significant scaling overhead and fails to capture fine-grained statistical affinity among weight values. To address this, we propose a data-driven partitioning strategy that clusters the unsalient set $\Omega_l^{\text{uns}}$ into $N_{\text{uns},l}$ disjoint groups, denoted $\{\Omega_l^{\text{uns}(k)}\}_{k=1}^{N_{\text{uns},l}}$, such that $\Omega_l^{\text{uns}} = \Omega_l \setminus \Omega_l^{\text{sal}} = \bigcup_{k=1}^{N_{\text{uns},l}} \Omega_l^{\text{uns}(k)}, \quad \Omega_l^{\text{uns}(i)} \cap \Omega_l^{\text{uns}(j)} = \varnothing$ for $i \neq j$. This clustering formulation enables group-specific quantization with shared scaling, reducing both error and storage overhead. To this end, we aim to determine a suitable clustering strategy and infer the optimal number of unsalient groups $N_{\text{uns},l}^{\text{optimal}}$ per matrix. While quantile-based partitioning aligns with Gaussian-like weight shapes, we observe (see appendix) that unsalient distributions vary significantly across layers—particularly early layers—which deviate from normality. As a result, fixed quantiles fail to capture this heterogeneity, degrading accuracy unless complemented by costly block-wise grouping. Unsupervised methods (e.g., K-means, Spectral clustering ) offer quality metrics but are impractical Zhang et al. (2025). For large unsalient sets $\Omega_l^{\text{uns}}$, such methods many clusters to avoid misassignments due to fine-grained variability and poor initialization.

We therefore propose a hybrid strategy that combines the statistical efficiency of quantile-based grouping with the adaptiveness of unsupervised clustering, We model the unsalient weight set $\Omega_l^{\text{uns}}$ as a graph $\mathcal{G}$, where nodes represent weights and edges encode pairwise affinities via Euclidean distance. Given the large size of LLM weight matrices, we apply randomized subsampling to select $S_l$ representative weights per layer, focusing on columns that capture attention patterns. A sparse $K$-nearest neighbor (KNN) graph is then constructed to form the affinity matrix, preserving the local structure of the unsalient distribution. To infer the optimal number of clusters, spectral clustering is applied to the affinity matrix across candidate values of cluster count. The number of clusters is constrained by the host memory budget, ensuring the average lookup bitwidth remains within the threshold $N_{\text{LUT}}$ (see Inference Runtime module). Cluster quality is evaluated using the Silhouette Score, which measures how well each weight fits within its assigned cluster, defined as:

$$s_r = \frac{1}{N} \sum_{i=1}^{N} \frac{v(i) - u(i)}{\max\{u(i), v(i)\}} \tag{2}$$

where $N$ is the number of nodes, $r$ the cluster count, $u(i)$ the average intra-cluster distance, and $v(i)$ the minimum average distance to other clusters. The optimal cluster count maximizes $s_r$, and the full unsalient grouping process is detailed in Algorithm 1 (Appendix). Now that the optimal number of unsalient clusters $N_{\text{uns},l}^{\text{optimal}}$ has been identified, we finalize the grouping by assigning weights to each subset $\Omega_l^{\text{uns}(k)}, \ k = 1, \ldots, N_{\text{uns},l}^{\text{optimal}}$, using a distribution-aware quantile scheme. Each subset is defined as $\Omega_l^{\text{uns}(k)} = \{w \in \Omega_l \mid \beta_l + z^{(k-1)}\gamma_l < |w| \leq \beta_l + z^{(k)}\gamma_l\}$, where $z^{(k)} = \Phi^{-1}\big((k \cdot \zeta_l^{\text{uns}} + 1)/2\big)$ and $\zeta_l^{\text{uns}} = 1 - \zeta_l^{\text{sal}}$.

## 3.3 MODULE 3: OPTIMIZED DUAL-MODE WEIGHT QUANTIZATION

We propose a dual-mode quantization method, where each mode targets a distinct subset of layer weights based on saliency. Given a weight matrix $W_l \in \mathbb{R}^{m \times n}$, we decompose it into a *salient component* $W_l^{\text{sal}} \in \mathbb{R}^{m \times n}$ and

multiple *unsalient components* $W_l^{\mathrm{uns}(k)} \in \mathbb{R}^{m \times n}$, each corresponding to a cluster of low-saliency weights:

$$W_l = W_l^{\mathrm{sal}} + \sum_{k=1}^{N_{\mathrm{uns},l}} W_l^{\mathrm{uns}(k)}, \tag{3}$$

$$\left(W_l^{\mathrm{sal}}\right)_{ij} = \begin{cases} w_{ij}, & \text{if } w_{ij} \in \Omega_l^{\mathrm{sal}}, \\ 0, & \text{otherwise}, \end{cases} \tag{4}$$

$$\left(W_l^{\mathrm{uns}(k)}\right)_{ij} = \begin{cases} w_{ij}, & \text{if } w_{ij} \in \Omega_l^{\mathrm{uns}(k)}, \\ 0, & \text{otherwise}. \end{cases} \tag{5}$$

Similar to Wang et al. (2025), we solve the quantization objective in Eq. 1 independently for each matrix component. Following similar approach to Wang et al. (2025) Mode 1 applies a multi-bit quantizer ($N_b$-bits) to the salient weights to preserve representational precision, while Mode 2 uses a 1-bit binarized quantizer for each unsalient group to maximize compression.

**Mode 1: Multibit Quantizer for Salient Weights**   The salient weight matrix $W_l^{\mathrm{sal}}$ is quantized into $\hat{W}_l^{\mathrm{sal}}$ using a channel-wise scaled multi-bit precision. To this end, we solve the general quantization formulation in Eq. 1, instantiated with a diagonal scaling matrix $A_l^{\mathrm{sal}} = \mathrm{diag}(a_1^{\mathrm{sal}}, \ldots, a_m^{\mathrm{sal}})$ and abinary-encoded matrix $B_l^{\mathrm{sal}} \in [-1, 1]^{m \times n}$, whose entries take values from a discrete set of size $2^{N_b}$ within $[-1, 1]$, with each value represented using $N_b$ binary bits. This reduces Equation 1 to the following optimization objective:

$$\min_{\mathbf{a}^{\mathrm{sal}}, B_l^{\mathrm{sal}}} \left\| W_l^{\mathrm{sal}} - \mathbf{a}^{\mathrm{sal}} \odot B_l^{\mathrm{sal}} \right\|_F^2, \qquad \hat{W}_l^{\mathrm{sal}} = \mathbf{a}^{\mathrm{sal}} \odot B_l^{sal} \tag{6}$$

where $\mathbf{a}^{\mathrm{sal}} = [a_1^{\mathrm{sal}}, \ldots, a_m^{\mathrm{sal}}]$ and $\odot$ denotes row-wise multiplication. The constraint ensures that $B_l^{\mathrm{sal}}$ is drawn from a finite set of $N_b$-bit quantized values in $[-1, 1]$. We analyze the convexity of Equation 6 by reformulating it row-wise to solve $\min_{a_i^{\mathrm{sal}}, b_{l,ij}^{\mathrm{sal}}} \| w_{l,ij}^{\mathrm{sal}} - a_i^{\mathrm{sal}} b_{l,ij}^{\mathrm{sal}} \|_2^2, \quad \forall j$. The problem is non-convex since each $b_{l,ij}^{\mathrm{sal}}$ can only take one of $2^{N_b}$ discrete values in $[-1, 1]$, making quantization a discrete optimization task. To address this, we relax the constraints on $B_l^{\mathrm{sal}}$ and allow its entries to take continuous values in $[-1, 1]$. We then apply an alternating optimization scheme: first solving a quadratic problem for $\mathbf{a}^{\mathrm{sal}}$, then updating $B_l^{\mathrm{sal}}$, and repeating until convergence. After convergence, the optimized $B_l^{\mathrm{sal}}$ is discretized by mapping values to the midpoints of $2^{N_b} + 1$ quantization levels. To better accommodate uneven distributions and tail-heavy outliers, the linear quantization levels $d \in [-1, 1]$ are further remapped using an exponential transformation:

$$r_{\mathrm{levels}}(d) = \mu_{B_l^{\mathrm{sal}}}^{\mathrm{opt}} + \sigma_{B_l^{\mathrm{sal}}}^{\mathrm{opt}} \cdot \mathrm{sign}(d) \cdot \left( \alpha \cdot \exp(|d|) - 1 \right), \tag{7}$$

where $\mu_{B_l^{\mathrm{sal}}}^{\mathrm{opt}}$ and $\sigma_{B_l^{\mathrm{sal}}}^{\mathrm{opt}}$ are statistics of the optimized $B_l^{\mathrm{sal}}$, and $\alpha$ controls the degree of nonlinearity. This remapping yields finer resolution for small magnitudes and coarser steps for outliers. The corresponding quantization centers $r_{\mathrm{centers}}$ are computed as midpoints of successive levels and used to discretize $B_l^{\mathrm{sal}}$.

**Mode 2: Binary Quantizer for Unsalient Weights**   Each unsalient component $W_l^{\mathrm{uns}(k)}$, $k = 1, \ldots, N_l^{\mathrm{uns,opt}}$, is quantized independently to $\hat{W}_l^{\mathrm{uns}(k)}$ using a quantizer. Each component is mapped to a ternary matrix $B_l^{\mathrm{uns}(k)} \in \{-1, 0, 1\}^{m \times n}$ and a single scalar $a_l^{\mathrm{uns}(k)} \in \mathbb{R}$ to minimize scaling overhead. Rewriting Eq. equation 1, we set $A = a_l^{\mathrm{uns}(k)} I_{m \times m}$ and $B = B_l^{\mathrm{uns}(k)}$, yielding the objective:

$$\min_{a_l^{\mathrm{uns}(k)}, B_l^{\mathrm{uns}(k)}} \left\| W_l^{\mathrm{uns}(k)} - a_l^{\mathrm{uns}(k)} B_l^{\mathrm{uns}(k)} \right\|_F^2, \quad \hat{W}_l^{\mathrm{uns}(k)} = a_l^{\mathrm{uns}(k)} \cdot B_l^{\mathrm{uns}(k)}. \tag{8}$$

To solve for $B_l^{\mathrm{uns}(k)}$, we apply an element-wise thresholding rule:

$$b_{l,ij}^{\mathrm{uns}(k)*} = \begin{cases} \mathrm{sign}(w_{l,ij}^{\mathrm{uns}(k)}) & \text{if } a_l^{\mathrm{uns}(k)} > 0, \\ -\mathrm{sign}(w_{l,ij}^{\mathrm{uns}(k)}) & \text{if } a_l^{\mathrm{uns}(k)} < 0, \\ 0 & \text{otherwise} \end{cases} \tag{9}$$

With $B_l^{\mathrm{uns}(k)}$ fixed, the optimal scalar is derived from the closed-form solution of Eq. 8:

$$a_l^{\mathrm{uns}(k)*} = \frac{\langle W_l^{\mathrm{uns}(k)}, B_l^{\mathrm{uns}(k)} \rangle}{\| B_l^{\mathrm{uns}(k)} \|_F^2}, \tag{10}$$

We denote $\langle W, B \rangle$ as the matrix inner product. While we refer to unsalient quantization as *binarization*, each $B_l^{\mathrm{uns}(k)}$ is ternary, taking values in $\{-1, 0, 1\}$, where zeros correspond to positions outside the $k$-th weight group. After full quantization, the overall binary matrix $B_l \in \{-1, 1\}^{m \times n}$ is constructed as: $B_l = \sum_{k=1}^{N_l^{\mathrm{uns,opt}}} B_l^{\mathrm{uns}(k)} + \mathrm{sign}(B_l^{\mathrm{sal}})$, $B_l^{\mathrm{uns}(k)} \in \{-1, 0, 1\}^{m \times n}$  The dual-mode quantization pipeline is summarized in Algorithm 2 in appendix.

## 3.4 MODULE 4: SALIENCY ALLOCATION VIA ADAPTIVE THRESHOLDING

Salient weights, which strongly deviate from the bulk distribution, are poorly suited for binarization. Misclassifying them as unsalient can lead to performance degradation Shang et al. (2023), while selecting too many reduces compression efficiency Wang et al. (2025). To balance compression and accuracy, we cast saliency allocation as a numerical optimization problem that adaptively selects the optimal salient percentile $\zeta_l^{\text{sal}}$ for each weight matrix $W_l$. The search is constrained by the salient bit budget, $N_b$, and $N_{\text{uns},l}^{\text{optimal}}$. The objective minimizes the normalized reconstruction error:

$$\min_{\zeta_l^{\text{sal}}} \mathcal{J}(\zeta_l^{\text{sal}}; W_l, N_{\text{uns},l}^{\text{optimal}}, N_b) = \frac{\|W_l^{\text{sal}} - \mathbf{a}^{\text{sal}} \odot B_l^{\text{sal}}\|_F^2 + \sum_{k=1}^{N_{\text{uns},l}^{\text{optimal}}} \|W_l^{\text{uns}(k)} - a_l^{\text{uns}(k)} B_l^{\text{uns}(k)}\|_F^2}{\|W_l\|_F^2},$$

$$\text{s.t.} \quad \zeta_l^{\text{sal}} \in [0, \zeta_l^{\text{sal,max}}]. \tag{11}$$

Here, $N_{\text{uns},l}^{\text{optimal}}$ is obtained from Algorithm 1, and the quantized matrices are computed using Algorithm 2 in appendix. Since the Frobenius norm is convex and the quantizers operate over relaxed representations of $B_l^{\text{sal}}$ and $B_l^{\text{uns}(k)}$, the objective $\mathcal{J}$ admits a global minimum. We solve this iteratively and efficiently using Brent's method Brent (2013), which combines golden-section search and parabolic interpolation for fast convergence. The final saliency threshold $\zeta_l^{\text{sal,opt}}$ guides the dual-mode quantization of $W_l$.

## 3.5 MODULE 5: EFFICIENT INFERENCE RUNTIME MODULE

We propose a scalable runtime group resolution mechanism that eliminates the need for costly block-wise saliency masks. Each weight matrix is paired with a *bit-packed* lookup table $L_l^{\text{LUT}}$, which encodes group indices for efficient routing to quantization scales during inference. For $N_{\text{uns},l}$ unsalient clusters and one salient group, this requires only: $N_{\text{LUT}} = \lceil \log_2(N_{\text{uns},l} + 1) \rceil$ bits per entry. The lookup table is stored in host memory and loaded layer-wise during inference. Because of its compact, bit-packed form, it introduces negligible overhead and enables loading of multiple layers metadata simultaneously, as allowed by device memory, thus reducing I/O latency. Although $L_l^{\text{LUT}}$ is not a traditional codebook, it functions as a decoding map for dynamic, group-aware quantization. This mechanism supports controllable trade-offs between storage cost and accuracy.

## 4 EXPERIMENTS

### 4.1 EXPERIMENTAL SETUP

We implement SAGE-PTQ in PyTorch Paszke et al. (2019) and Huggingface Wolf et al. (2019), running all experiments on a single NVIDIA A100 80GB GPU. Models up to 30B parameters can also run on an NVIDIA L40 46GB GPU. SAGE-PTQ operates directly on pretrained models without fine-tuning

**Models and Datasets** We apply SAGE-PTQ to LLaMA Touvron et al. (2023a), LLaMA-2 Touvron et al. (2023b), and OPT Zhang et al. (2022) models up to 70B parameters, and extend evaluation to diverse architectures including Vicuna Chiang et al. (2023), DeepSeek 7B AI (2024), and LLaMA-3-8B Dubey et al. (2024). Performance is measured using perplexity, a standard metric in quantization studies, on WikiText-2 Merity et al. (2016), PTB Marcus et al. (1994), and a subset of C4 Raffel et al. (2020). Perplexity is chosen for its sensitivity to distributional shifts introduced by compression. To assess generalization, we also evaluate zero-shot classification accuracy on six NLS benchmarks: PIQA Bisk et al. (2020), BoolQ Clark & Lee (2019), OpenBookQA Mihaylov et al. (2018), Winogrande Sakaguchi et al. (2021), ARC-c Clark & Etzioni (2018), and HellaSwag Zellers et al. (2019). Compared to their FP16 counterparts, quantized models retain high performance, with SAGE-PTQ maintaining accuracy across tasks and model families with minimal degradation.

**Baseline** Our main baseline is BiLLM Huang et al. (2024), the SoTA in binarized LLMs, which adopts column-wise saliency selection and unsalient weight grouping. We also include PB-LLM Shang et al. (2023), which introduced element-wise saliency-based binarization, and GPTQ Frantar et al. (2023) under 2-bit quantization, as both BiLLM and PB-LLM build on the GPTQ framework.

### 4.2 RESULTS

**Evaluation of Language Generation Tasks** We evaluate SAGE-PTQ across multiple LLMs and sizes, comparing against BiLLM Huang et al. (2024) and other sub-2-bit methods Shang et al. (2023); Frantar et al. (2023). Perplexity serves as the main metric for generation quality. We also assess binarized precision and explicitly account for scaling overhead during inference, an aspect often neglected in prior work.

Table 1: Perplexity results of GPTQ, PB-LLM (10%), BiLLM and SAGE-PTQ on the OPT family. The columns represent the perplexity results on WikiText2 datasets with different model sizes. SAGE-PTQ achieves superior binarization performance with minimal scale overhead.

| Method | Weight bits | Scale bits | 1.3B | 2.7B | 6.7B | 13B | 30B | 66B |
|---|---|---|---|---|---|---|---|---|
| GPTQ | 2.00 | 0.25 | 115.17 | 61.59 | 50.19 | 21.36 | 15.71 | 82.10 |
| PB-LLM (10%) | 1.70 | 0.50 | 265.52 | 124.35 | 105.16 | 81.92 | 25.14 | 29.09 |
| BiLLM | 1.11 | 1.00 | 69.97 | 49.55 | 35.36 | 18.82 | 12.71 | 12.06 |
| **SAGE-PTQ ($N_{LUT}$=3)** | **1.07** | **0.009** | **17.90** | **14.60** | **13.87** | **11.33** | **10.09** | **11.35** |
| **SAGE-PTQ ($N_{LUT}$=4)** | **1.07** | **0.009** | **15.63** | **13.23** | **11.59** | **10.91** | **9.82** | **11.31** |

Table 2: WikiText2 perplexity results of GPTQ, PB-LLM (10%), BiLLM and SAGE-PTQ on LLaMA-1 and LLaMA-2. SAGE-PTQ consistently achieves lower perplexity and scale overhead. *: LLaMA has a 65B variant; LLaMA-2 has a 70B variant.

| Model | Method | Weight bits | Scale bits | 7B | 13B | 30B | 65B / 70B* |
|---|---|---|---|---|---|---|---|
| **LLaMA-1** | GPTQ | 2.00 | 0.25 | 152.31 | 20.44 | 13.01 | 8.78 |
| | PB-LLM (10%) | 1.70 | 0.50 | 102.36 | 36.60 | 33.67 | 12.53 |
| | BiLLM | 1.09 | 1.00 | 35.04 | 15.14 | 10.52 | 8.49 |
| | **SAGE-PTQ** | **1.03** | **0.004** | **5.99** | **5.35** | **5.97** | **4.93** |
| **LLaMA-2** | GPTQ | 2.00 | 0.25 | 60.45 | 19.70 | NA | 9.12 |
| | PB-LLM (10%) | 1.00 | 0.50 | 69.20 | 151.09 | NA | 28.37 |
| | BiLLM | 1.08 | 1.00 | 32.48 | 16.77 | NA | 8.41 |
| | **SAGE-PTQ** | **1.03** | **0.004** | **5.87** | **14.49** | **NA** | **7.51** |

Table 1 presents SAGE-PTQ results on OPT models from 1.3B to 66B using 2-bit precision for salient weights. To mitigate degradation from outlier-heavy first layer matrices (Appendix), we allocate 4-bit precision to their salient weights. Experiments are conducted with lookup constraints $N_{LUT} = 3$ and $N_{LUT} = 4$, comparing WikiText-2 perplexity against BiLLM, PB-LLM, and GPTQ. The 4-bit setting matches baseline storage, while the 3-bit setting highlights robustness under tighter constraints. Recall that lookup values are decoded once per layer and do not add to device memory overhead. SAGE-PTQ achieves the best perplexity across all model sizes and lookup constraints. While enforcing a maximum of 10% salient weights, our adaptive saliency selection yields an average of 7%, resulting in a binarized precision of 1.07. Crucially, SAGE-PTQ requires only **0.009 bits per weight** for scale storage, far below all competing methods. This scaling overhead is computed by accounting for the number of FP16 scale values required to decode weights at inference. GPTQ and PB-LLM use block-wise min-max scaling on 128-column blocks. GPTQ stores one min and one max column (2 FP16 columns), contributing 0.25 bits per weight. PB-LLM stores 4 columns (2 for salient, 2 for unsalient), totaling 0.5 bits. BiLLM further adds residual scaling and uses 8 columns per block (4 for salient, 4 for unsalient), requiring 1.0 bit. In contrast, SAGE-PTQ uses a single FP16 scale for the entire matrix and one scalar per unsalient cluster, avoiding position-based heuristics and yielding negligible scale cost.

We evaluate SAGE-PTQ on LLaMA-1 and LLaMA-2 up to 70B, under lookup constraint $N_{LUT} = 4$, results are shown in Table 2. For models under 30B, only 1% of weights are marked salient and quantized with 4-bit precision in all layers due to outlier sensitivity. SAGE-PTQ achieves a 1.03 average bitwidth and 0.004 bits of scaling overhead, outperforming all baselines in perplexity.

We evaluate SAGE-PTQ with $N_{LUT} = 4$ on six 7B-scale models: LLaMA-(1,2,3)-7B, OPT-6.7B, Vicuna-7B (instruction-tuned), and DeepSeek-7B (efficient architecture),across datasets WikiText2, PTB, and C4. Figure 2 show that SAGE-PTQ consistently outperforms BiLLM across all WikiText2, PTB, and C4, reducing perplexity by 77.6%, 87.3%, and 74.2%, respectively. LLaMA and Vicuna benefit from 4-bit saliency at a 1% threshold, while OPT and DeepSeek achieve strong results with 2-bit saliency at 10%, due to broader weight variance and fewer outliers. We resume more experimental analysis of larger model sizes and different architectures across the three datasets, results are in the appendix.

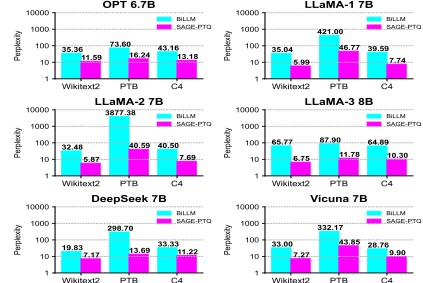

Figure 2: Perplexity comparison of SAGE-PTQ and BiLLM on various 7B-scale models. Reported on WikiText2, PTB, and C4 datasets.

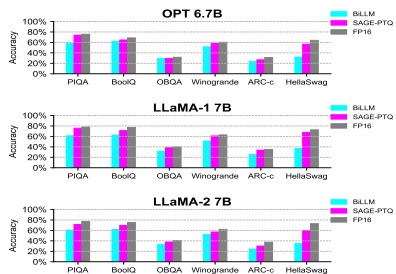

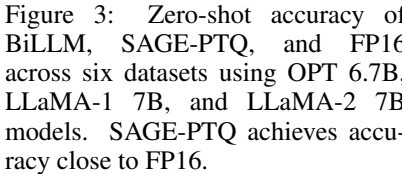

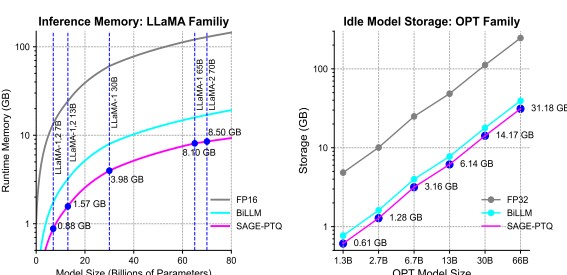

Figure 3: Zero-shot accuracy of BiLLM, SAGE-PTQ, and FP16 across six datasets using OPT 6.7B, LLaMA-1 7B, and LLaMA-2 7B models. SAGE-PTQ achieves accuracy close to FP16.

Figure 4: Memory footprint comparison between SAGE-PTQ and BiLLM for LLaMA and OPT model families. SAGE-PTQ uses less than 50% of BiLLM's GPU memory and achieves lower storage requirements in idle mode.

**Zero-shot Results** We evaluate SAGE-PTQ under lookup constraints ($N_{\text{LUT}} = 3$) on six zero-shot NLU QA benchmarks: PIQA, BoolQ, OBQA, Winogrande, ARC-e, and HellaSwag, using LLaMA-1 7B, LLaMA-2 7B, and OPT-6.7B. SAGE-PTQ matches FP16 accuracy on most tasks, outperforming BiLLM by an average of 5% and staying within 4% of FP16 across all benchmarks.

**Memory Footprint** A key motivation for LLMs PTQ is reducing GPU memory usage during inference for edge deployment. We evaluate the memory footprint of SAGE-PTQ, which, unlike prior methods Huang et al. (2024); Shang et al. (2023), avoids untracked overhead by offloading lookup metadata to host memory, requiring only weights and scales on GPU. We follow standard practice by comparing against FP16 for active GPU inference and FP32 for idle disk storage. For BiLLM comparison, we assume best-case lookup overhead, as its group retrieval mechanism is unspecified. Following prior work Shang et al. (2023); Frantar et al. (2023), BiLLM uses block-wise grouping: 32 blocks of size 128 for 7B models and up to 64 blocks for 70B. Assuming fixed block positions and three groups per block, we use a best-case estimate of $\lceil \log_2(3) \rceil = 2$ bits per weight matrix for lookup storage, included alongside weight and scale bits in total storage. We measure runtime GPU memory for LLaMA-1 and LLaMA-2, and idle host memory for OPT. Scale bit values are from Tables 1 and 2. As shown in Figure 4, SAGE-PTQ achieves SoTA efficiency, averaging 1.038 bits per weight including scaling. In contrast, BiLLM requires 2.08 bits, resulting in a 50.3% reduction in GPU memory. For OPT, we evaluate with $N_{\text{LUT}} = 4$ bits; even under this constraint, SAGE-PTQ lowers idle storage by 20.7%.

**End-to-end Inference Latency** We report the inference latency of SAGE-PTQ through a case study on LLaMA-2-70B (140 GB in FP16) deployed on a single NVIDIA L40 GPU (46 GB). Since the model has 80 layers and cannot fully reside in GPU memory, standard FP16 deployment loads a small band of layers (e.g., $B_{FP16} = 5$) from CPU RAM per token, introducing significant I/O latency. With SAGE-PTQ and a lookup budget of $N_{\text{LUT}} = 3$, the quantized model fits in $\sim 12.7$ GB, with $\sim 3$ GB for KV-cache during prefill for a context length of 2048. During decoding, SAGE-PTQ enables on-the-fly dequantization of a larger band ($B_Q = 16$) using compact lookup metadata, avoiding full layer transfers. We measure both layer preprocessing time and per-token generation latency in table 3, highlighting how SAGE-PTQ introduces moderate dequantization latency but offers the potential for overall inference speedup by reducing I/O bottlenecks. Results show that SAGE-PTQ introduces moderate dequantization overhead, but significantly reduces I/O latency by avoiding full layer transfers. By enabling in-GPU preprocessing and parallel dequantization of bands, it achieves a 1.5× speedup in per-token decoding time of LLaMA-2-70B, making it well-suited for memory-constrained edge-deployed scenarios. For a given a context length $L_0$, band I/O overhead $d_Q$, and dequantization cost $D$, the per-token latency can be modeled to capture the trade-off between memory efficiency and latency. For generating short sequences of length $T \ll L_0$, the total latency for the quantized model becomes: $\approx O\left(T \cdot \frac{N_L}{B_Q}(d_Q + D)\right)$ demonstrating approximately linear scaling with sequence length $T$ and hardware overheads.

Table 3: Inference Latency Breakdown for LLAMA-2-70B Quantized with SAGE-PTQ

| Method | Layer I/O Overhead | Band Dequantization | Per-token Latency |
|---|---|---|---|
| FP16 | 47.07 ms | – | 972.1 ms |
| **SAGE-PTQ** | 10.46 ms | 75.52 ms | **646.3 ms** |

Table 4: Runtime for full quantization of OPT family. **SAGE-PTQ** achieves faster quantization.

| Method | 1.3B | 2.7B | 6.7B | 13B | 30B | 66B |
|---|---|---|---|---|---|---|
| BiLLM | 5.1m | 8.9m | 15.8m | 26.4m | 51.9m | 1.7h |
| **SAGE-PTQ** | **4.2m** | **6.6m** | **10.2m** | **19.5m** | **43.9m** | **1.5h** |

**Quantization Time** We evaluate the quantization efficiency of the proposed SAGE-PTQ modules by measuring the total time required to quantize full models on a single NVIDIA L40 GPU. As shown in Table 4, SAGE-PTQ consistently achieves faster quantization times compared to BiLLM, despite involving multiple specialized modules. Notably, it is capable of quantizing a 66B parameter model ($\sim$ 140GB) in just 1.5 hours, highlighting efficient device-optimized implementation.

## 4.3 ABLATION STUDY

SAGE-PTQ has five-module design that enables binarized inference by combining adaptive saliency selection and graph-based unsalient weight clustering, we evaluate the effectiveness of these components. We also evaluate its efficiency in balancing memory footprint and performance. More analysis are discussed in Appendix.

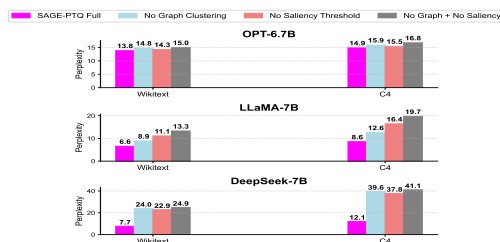

Figure 5: Ablation results of SAGE-PTQ on OPT 6.7B, LLaMa 7B, and DeepSeek-base 7B. Full approach achieves best perplexity.

Figure 6: Perplexity on Wikitext2 and C4 under different lookup constraints on OPT family. 4-bit Lookup resolution balances performance-storage tradeoff

**Effectiveness of Saliency Thresholding and Graph-based Modeling** We ablate Modules 2 (adaptive saliency thresholding) and 4 (graph-based inlier grouping) in SAGE-PTQ, evaluating four variants on WikiText2 and C4 using LLaMA-7B, OPT-6.7B, and DeepSeek-7B, chosen for architectural and weight distribution diversity. Module 3 (quantization engine) is fixed. Figure 5 shows removing either module increases perplexity. The full method yields the lowest perplexity, improving 2.4× on outlier-heavy models, LLaMA and Deepseek-base, and showing consistent gains on OPT 6.7B, model with regular weight distributions.

**Tradeoff between Lookup Overhead and Quantization Performance** SAGE-PTQ Employs a novel lookup strategy to recover weight cluster information at inference, we evaluate the effect of varying the number of weight clusters on model performance. As shown in Figure 6, experiments on the OPT family demonstrate that increasing the lookup size improves perplexity on both WikiText2 and C4, due to finer granularity and more accurate scale assignment. However, even with a 4-bit lookup, the method closely matches the performance of higher settings, indicating its ability to capture salient weight structure without requiring high-resolution overhead.

## 5 CONCLUSION

In this work, we introduced SAGE-PTQ, a novel PTQ framework for LLMs that addresses key limitations of prior binarization methods. Our modular design jointly optimizes saliency selection, weight grouping, and precision allocation, enabling significant gains in perplexity and inference efficiency across diverse LLM families and scales. Extensive experiments show that SAGE-PTQ consistently outperforms SoTA baselines such as BiLLM and PB-LLM. On language modeling tasks, it achieves up to 5× lower perplexity while using just 1.03 average bits per weight, with only 0.004 bits of scaling overhead. Compared to BiLLM, SAGE-PTQ reduces runtime GPU memory usage by over 50% on the LLaMA family. It also preserves full-precision zero-shot NLU accuracy, with less than 4% degradation. While effective, our method currently lacks adaptive precision-aware allocation for outlier weights. Nonetheless, our results establish SAGE-PTQ as a strong candidate for deploying LLMs on edge and resource-constrained platforms. SAGE-PTQ code will be released soon.

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

## A  MODULE 2 AND MODULE 3 ALGORITHMS

In this section, we detail the key algorithms in SAGE-PTQ, focusing on Module 2, which determines the optimal number of unsalient groups per weight matrix, and Module 3, which implements the dual-mode quantization pipeline.

Module 2 discovers the optimal number of unsalient weight clusters $N_{\text{uns},l}^{\text{optimal}}$ using spectral clustering guided by a silhouette-based scoring strategy. If the number of unsalient weights $|\Omega_l^{\text{uns}}|$ exceeds the sampling budget $S_l$, a random subset is selected for efficiency. A sparse $K$-nearest neighbor graph is constructed over the subset, and spectral clustering is performed for each candidate number of clusters $r \in [2^{N_{\text{LUT}}-1} + 1, 2^{N_{\text{LUT}}} - 1]$. The silhouette score is computed for each $r$, and the clustering configuration that maximizes this score is chosen as the optimal grouping.

Module 3: Dual-Mode Quantization applies distinct quantization strategies to the salient and unsalient components of each weight matrix. For salient weights, the method uses a multi-bit quantization procedure that iteratively optimizes per-channel scaling factors and normalizes the weights. These are then quantized using an adaptive level selection strategy, which assigns values based on learned quantization centers, minimizing quantization error. In contrast, each unsalient group is binarized using a group-wise binary matrix and a single scalar scaling factor, resulting in highly efficient compression. This dual-mode design ensures that the quantization process preserves critical model behavior through fine-grained treatment of salient weights while maximizing efficiency by aggressively compressing the less informative weights.

## B  STATISTICAL ANALYSIS OF WEIGHT MATRICES AND OUTLIER PATTERN DETECTION

Efficient post-training quantization for large language models (LLMs) demands careful modeling of weight matrix statistics, including *saliency*, *sparsity*, and *outliers*. SAGE-PTQ is a saliency-aware framework designed from empirical analysis across model families and layers. It enables *mixed-precision quantization* by preserving higher precision for salient weights while reducing overall bit usage. This data-driven design ensures adaptability to diverse architectures and yields improved trade-offs between compression and accuracy across LLMs.

We begin by analyzing the OPT model family, focusing on OPT-6.7B as a representative intermediate-scale model with 32 transformer layers. For each layer, we study the value distribution of self-attention projection weight matrices—*query*, *key*, *value*, and *output*—each of size $4096 \times 4096$. We track how these distributions evolve from early (layer 0) to mid and final layers (e.g., layer 30). Similar analyses are conducted on LLaMA-1 7B and DeepSeek-Base 7B to compare distributional patterns across model families. For each weight matrix, we compute histograms of individual values and fit a Gaussian curve using the empirical mean and standard deviation. We calculate the Kullback–Leibler (KL) divergence between the histogram and fitted Gaussian to

---

**Algorithm 1** Module 2: Unsalient Group Count Discovery

---

1: **Input:** $\Omega_l^{\text{uns}}$, $N_{\text{LUT}}$, $S_l$, $K_{\text{neighbors}}$
2: **Output:** $N_{\text{uns},l}^{\text{optimal}}$
3: **if** $|\Omega_l^{\text{uns}}| > S_l$ **then**
4:     $\Omega_{\text{sub}} \leftarrow$ Random subset of $\Omega_l^{\text{uns}}$ with size $S_l$
5: **else**
6:     $\Omega_{\text{sub}} \leftarrow \Omega_l^{\text{uns}}$
7: **end if**
8: Construct sparse $K_{\text{neighbors}}$-NN graph $A$ over $\Omega_{\text{sub}}$
9: best_score $\leftarrow -\infty$, $N_{\text{uns},l}^{\text{optimal}} \leftarrow 0$
10: **for** $r = 2^{N_{\text{LUT}}-1} + 1$ to $2^{N_{\text{LUT}}} - 1$ **do**
11:     Perform spectral clustering on $\mathcal{G}$ with $r$ clusters
12:     Compute silhouette score $s_r$:
13:     **if** $s_r > $ best_score **then**
14:         best_score $\leftarrow s_r$,    $N_{\text{uns},l}^{\text{optimal}} \leftarrow r$
15:     **end if**
16: **end for**
17: **return** $N_{\text{uns},l}^{\text{optimal}}$

---

---

**Algorithm 2** Dual-Mode Quantization for Salient and Unsalient Weight Components

---

**Require:** $W_l^{\text{sal}}, \{W_l^{\text{uns}(k)}\}_{k=1}^{N_{\text{uns},l}^{\text{optimal}}}, N_b$, iters

**Ensure:** Quantized matrices $\hat{W}_l^{\text{sal}}, \{\hat{W}_l^{\text{uns}(k)}\}_{k=1}^{N_{\text{uns},l}^{\text{optimal}}}$

1: **Mode 1: Quantize Salient Component**

2: Initialize $B_l^{\text{sal}} \leftarrow \text{sign}(W_l^{\text{sal}}), \mathbf{a}^{\text{sal}}{}^{\text{sal}} \leftarrow \mathbf{0}$

3: **for** iter = 1 to iters **do**

4:     Update scaling: $a_i^{\text{sal}} \leftarrow \frac{\sum_j W_{ij}^{\text{sal}} B_{ij}^{\text{sal}}}{\sum_j (B_{ij}^{\text{sal}})^2}$

5:     Clip and normalize: $B_l^{\text{sal}} \leftarrow \text{clip}\left(\frac{W_l^{\text{sal}}}{\mathbf{a}^{\text{sal}}{}^{\text{sal}}[:,\text{None}]}, -1, 1\right)$

6: **end for**

7: **Adaptive Quantization Step:**

8: Compute $\mu_B, \sigma_B$ from nonzero elements of $B_l^{\text{sal}}$

9: Compute quantization levels using equation 7

10: $r_{\text{centers}} \leftarrow \frac{r_{\text{levels}}[:-1] + r_{\text{levels}}[1:]}{2}$

11: $B_l^{\text{sal}} \leftarrow r_{\text{centers}}[\arg\min |B_l^{\text{sal}} - r_{\text{centers}}|]$

12: $\hat{W}_l^{\text{sal}} \leftarrow \mathbf{a}^{\text{sal}}{}^{\text{sal}} \odot B_l^{\text{sal}}$

13: **Mode 2: Binarize Each Unsalient Component**

14: **for** each $k = 1$ to $N_{\text{uns},l}^{\text{optimal}}$ **do**

15:     Compute binary matrix $B_l^{\text{uns}(k)}$ using equation 9

16:     Compute scalar $a_l^{\text{uns}(k)}$ using equation 10

17:     $\hat{W}_l^{\text{uns}(k)} \leftarrow a_l^{\text{uns}(k)} \cdot B_l^{\text{uns}(k)}$

18: **end for**

19: **return** $\hat{W}_l^{\text{sal}}, \{\hat{W}_l^{\text{uns}(k)}\}_{k=1}^{N_{\text{uns},l}^{\text{optimal}}}$

---

quantify deviation from normality. This tests whether the law of large numbers (LLN) applies and reveals layers with statistical anomalies or salient outliers. These observations guide the design of our quantization method, ensuring adaptability to model-specific and layer-wise weight characteristics.

**OPT-6.7B:**   As shown in figure 7, Most layers in the OPT-6.7B model exhibit weight distributions that closely match a Gaussian curve, with KL divergence typically below 0.1. The long tail of the histogram contains salient weights, and the tail length varies across layers. The first layer deviates significantly from Gaussianity, suggesting early-layer adaptation. Standard deviation of weights varies across layers, indicating nonuniformity in weight distribution statistics.

**LLaMA-1 7B:**   In figure 8, The LLaMA-1 7B model displays a small number of high-magnitude outliers. While the bulk of the weight distribution remains approximately Gaussian, sharp deviations are again observed in the first layer. The sensitivity of outliers to quantization and the variation in standard deviation across layers suggest the need for adaptive, outlier-aware weight grouping.

**DeepSeek-7B:**   Figure 9 shows that the DeepSeek-7B model consistently deviates from Gaussianity across all layers. It contains high-magnitude outliers in long-tailed distributions. Inlier weights exhibit relatively small but layer-dependent standard deviations. These observations suggest that fewer weight groups are required, with more bits allocated to salient weights.

- OPT-6.7B layers are mostly Gaussian, except for strong deviation in the first layer.

- LLaMA-1 7B contains large-magnitude outliers and early-layer deviations.

- DeepSeek-7B shows persistent non-Gaussian behavior and long-tailed outliers.

- Standard deviation of inlier weights varies significantly across layers.

- Inlier weight distributions are layer-specific and inconsistent across models.

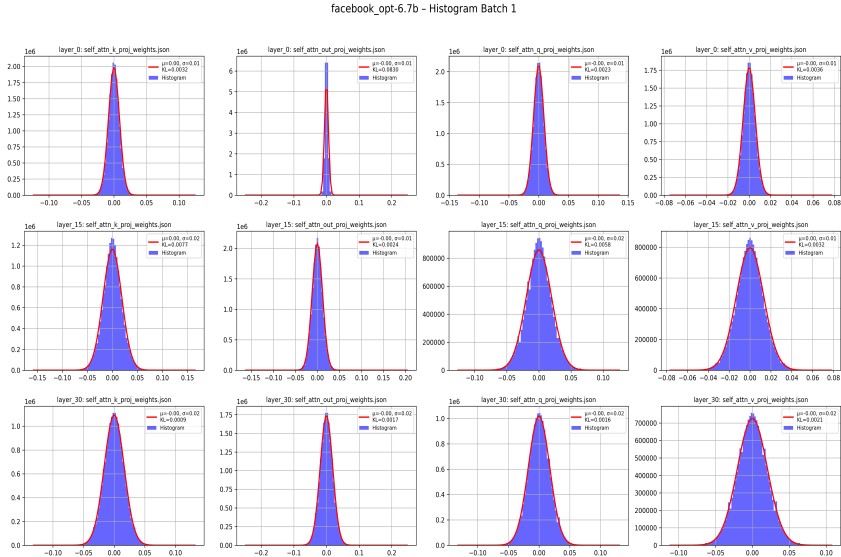

Figure 7: Distribution of self-attention projection layer weights in OPT-6.7B across selected layers (0,15,30). All layers follow a near-Gaussian trend with low KL divergence, except for the out_proj matrix in the first layer, which shows significant deviation.

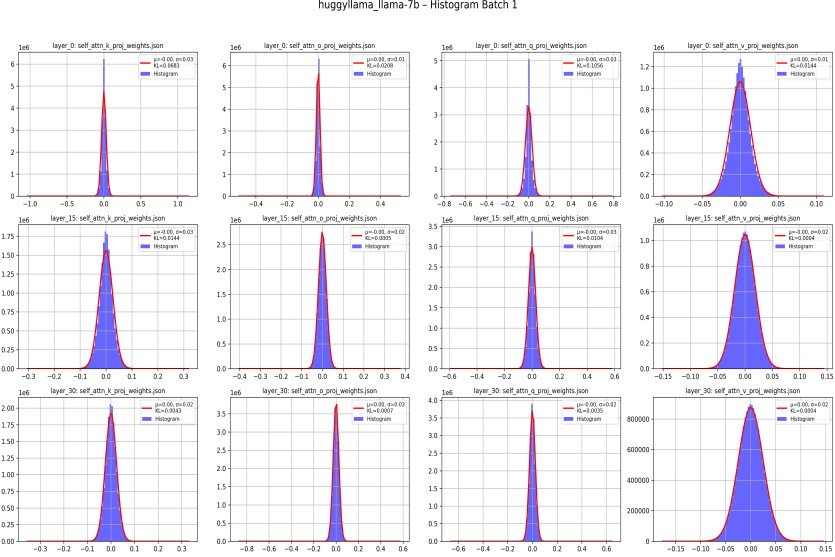

Figure 8: Distribution of self-attention projection layer weights in LLaMa-1 7B across Self-Attention selected layers (0,15,30). Observed Gaussian Trends with Sparse High-Magnitude Outliers.

**Quantization Strategy Requirements:**

- Detect and isolate outliers before quantization.
- Avoid fixed-position or global threshold-based clustering.
- Assign higher precision to salient weights to prevent quantization distortion.
- Use affinity-based grouping for inlier weights instead of value range alone.
- Apply a graph-based clustering method to model intra-layer similarity.

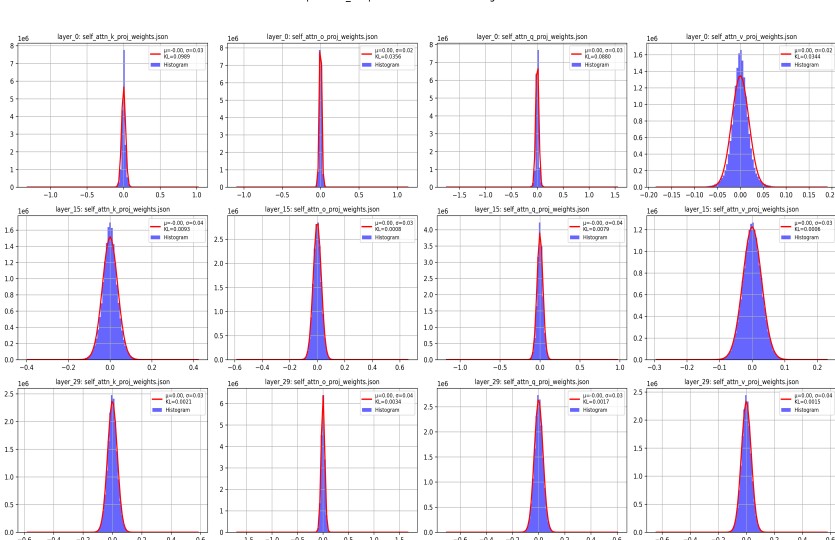

Figure 9: Self-Attention Weight Distributions in DeepSeek-7B of selected layers (0,15,29): Non-Gaussian Patterns with Widespread High-Magnitude Outliers.

- Adapt group count and saliency thresholds dynamically per layer.

These insights motivated the design of SAGE-PTQ, including:

- **Module 1**: Saliency filtering to isolate high-impact weights,
- **Module 2**: Graph-based clustering to group inliers by affinity,
- **Module 4**: Adaptive saliency thresholding to prevent outlier contamination within inlier groups.

# C    EXTENDED EXPERIMENTAL RESULTS

## C.1    LANGUAGE GENERATION TASKS

To rigorously evaluate the generalization and robustness of SAGE-PTQ across diverse model families and scales, we extend our analysis to larger models beyond the 7B parameter range. We report perplexity on three benchmark datasets: WikiText2, PTB, and C4, and compare our results against BiLLM. We only compare against BiLLM since it achieves strong performance at comparable bitwidths, particularly in binarized settings.

All models are evaluated under a lookup table constraint of $N_{\text{LUT}} = 4$ bits. For OPT models, which exhibit regular Gaussian weight distributions with consistent outlier behavior (as detailed in Appendix A), we assign $N_b = 2$ bits to salient weights and $N_b = 4$ for the first layer to account for its deviation from normality. A saliency threshold of 10% is used. For LLaMA and Vicuna models, where outliers are rare but extreme, we apply a stricter 1% saliency threshold. SAGE-PTQ achieves an average bitwidth of 1.07 for OPT and 1.03 for LLaMA/Vicuna, outperforming BiLLM on all datasets. Our method delivers new state-of-the-art results with perplexity scaling factors as low as 0.004 for LLaMA/Vicuna and 0.009 for OPT.

## C.2    EVALUATION OF SALIECY METRIC

We conducted additional analysis on SAGE-PTQ to compare calibration-based saliency metrics used for identifying quantization-sensitive weights. Specifically, we evaluated the classical Hessian-based sensitivity metric adopted in prior work such as BiLLM, defined as $S_{ij} = \frac{W_{ij}^2}{H_{ii}^{-1}}$, where $H_{ii}^{-1}$ is approximated using Cholesky decomposition of the activation Gram matrix. Our goal was to determine whether second-order statistics are necessary for saliency detection, or if our affinity-based strategy—focused on preserving weight statistics and minimizing quantization error—is more effective. We also evaluated a strategy where all weights, including salient ones, are binarized uniformly to test the importance of preserving high-precision weights. As shown in Figure 11, the magnitude-based saliency metric outperforms all others across different models and datasets.

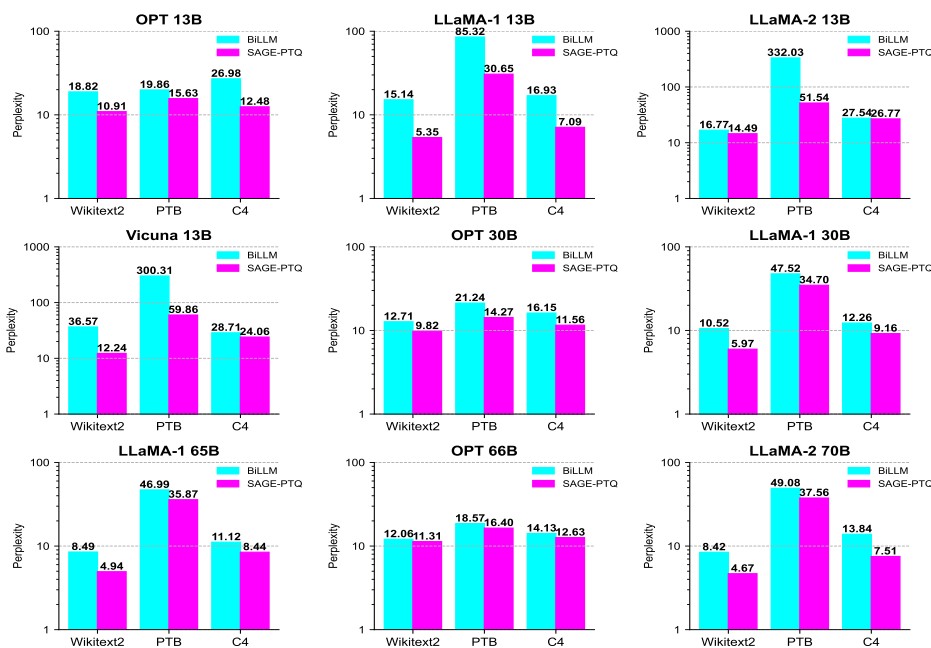

Figure 10: Extended evaluation of SAGE-PTQ versus BiLLM under a lookup table constraint of $N_{\text{LUT}} = 4$ bits across multiple model families (OPT, LLaMA-1, LLaMA-2, and instruction-tuned Vicuna) ranging from 13B to 70B parameters. The bar plots report perplexity on WikiText2, PTB, and C4 datasets. SAGE-PTQ consistently outperforms BiLLM across all models, achieving an average precision of 1.03–1.07 bits per weight.

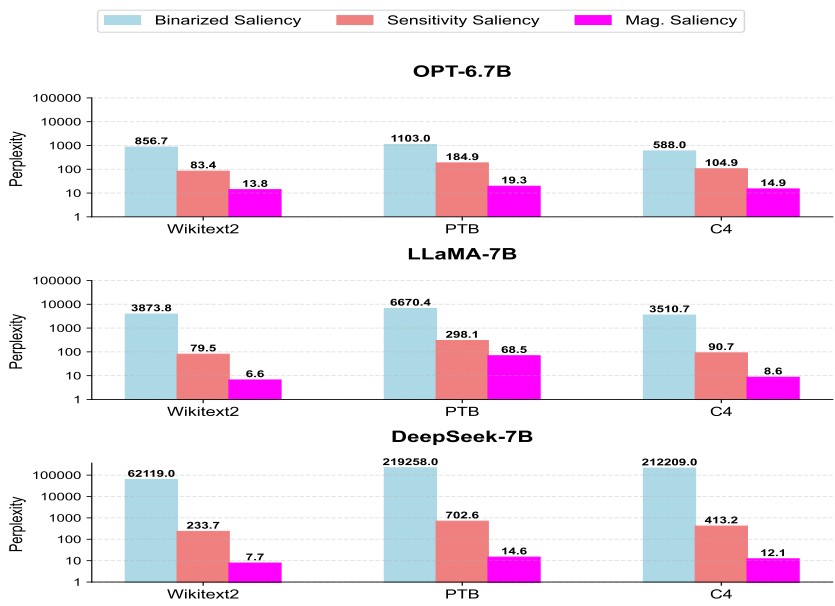

Figure 11: SAGE-PTQ Ablation study on saliency metrics across three model families (OPT-6.7B, LLaMA-7B, DeepSeek-7B) and three datasets (WikiText2, PTB, C4). We compare perplexity results under three conditions: (1) magnitude-based saliency, (2) Hessian-based saliency, and (3) uniform binarization of all weights without saliency distinction. Magnitude-based saliency consistently yields the lowest perplexity, demonstrating its effectiveness and efficiency for SAGE-PTQ method

Table 5: Comparison of quantization methods across three large-scale models. SAGE-PTQ achieves the lowest GPU occupancy and perplexity despite using binary weights and negligible scaling overhead.

| Model | Metric | GPTQ | PB-LLM (10%) | BiLLM | SAGE-PTQ |
|---|---|---|---|---|---|
| OPT 66B | Weight Bits | 2.00 | 1.70 | 1.11 | **1.07** |
| | Scale Bits | 0.25 | 0.50 | 1.00 | **0.009** |
| | Block Size | 128 | 128 | 128 | **N/A** |
| | GPU Occupancy | 14.06% | 13.75% | 13.19% | **6.70%** |
| | Wikitext2 Perplexity | 82.10 | 29.09 | 12.06 | **11.31** |
| LLaMA-1 65B | Weight Bits | 2.00 | 1.70 | 1.09 | **1.03** |
| | Scale Bits | 0.25 | 0.50 | 1.00 | **0.004** |
| | Block Size | 128 | 128 | 128 | **N/A** |
| | GPU Occupancy | 14.06% | 13.75% | 13.06% | **6.46%** |
| | Wikitext2 Perplexity | 8.78 | 12.53 | 8.49 | **4.93** |
| LLaMA-2 70B | Weight Bits | 2.00 | 1.70 | 1.08 | **1.03** |
| | Scale Bits | 0.25 | 0.50 | 1.00 | **0.004** |
| | Block Size | 128 | 128 | 128 | **N/A** |
| | GPU Occupancy | 14.06% | 13.75% | 13.00% | **6.46%** |
| | Wikitext2 Perplexity | 9.12 | 28.37 | 8.41 | **4.67** |

Uniform binarization of salient weights leads to substantial degradation, confirming their impact on model performance. However, Hessian-based metrics did not yield a significant advantage over simple magnitude, indicating that magnitude alone is a reliable and efficient saliency indicator.

### C.3 RUNTIME EFFICIENCY

A key contribution of the SAGE-PTQ method is its ability to reduce scaling overhead during inference to a negligible fraction relative to the size of the weight matrices, while supporting true binary precision. To demonstrate this efficiency, we compare SAGE-PTQ with SoTA ultra-low precision baselines: GPTQ (2-bit), PB-LLM (sub-2-bit), and BiLLM (binary), across three large-scale models—OPT-66B, LLaMA-1 65B, and LLaMA-2 70B. Our evaluation includes all untracked scaling overheads in each method. Results show that SAGE-PTQ achieves the lowest inference-time memory footprint, requiring less than 50% of the memory used by the most efficient baseline (BiLLM). Moreover, despite operating at binary precision with minimal overhead, our method outperforms all baselines in Wikitext2 perplexity without relying on position-based heuristics, as summarized in Table 5.

## D IMPACT OF LOOKUP TABLE SIZE ON PERFORMANCE

Quantization resolution plays a critical role in controlling the loss induced by post-training quantization. In SAGE-PTQ, resolution is determined by the number of weight groups, each associated with a unique scaling factor. Increasing the number of groups improves quantization fidelity but requires a larger lookup table (LUT) to store group indices for inference. While SAGE-PTQ ensures that scale overhead remains negligible—requiring only one scalar per group—lookup size introduces a trade-off between performance and idle storage.

We evaluate the effect of varying lookup sizes (3 to 6 bits, i.e., up to 64 groups) on perplexity across three model families: OPT (1.3B to 66B), LLaMA-1 and LLaMA-2 (7B to 70B), and instruction-tuned Vicuna (7B, 13B). Experiments are conducted on Wikitext2, PTB, and C4 datasets.

As shown in Figure 12, a 3-bit lookup (8 groups) yields suboptimal performance. However, beyond 4 bits, the perplexity gain saturates. For OPT models, which exhibit stable weight distributions (see Appendix A), even 3-bit lookups suffice. In contrast, LLaMA models benefit from 4-bit lookups, consistent with their slight deviation from Gaussianity. Instruction-tuned models like Vicuna show notable gains with higher LUT sizes, likely due to fine-tuning introducing specialized weight distributions.

We select a 4-bit LUT as the default setting, offering a good trade-off between accuracy and storage, and enabling fair comparison with prior methods.

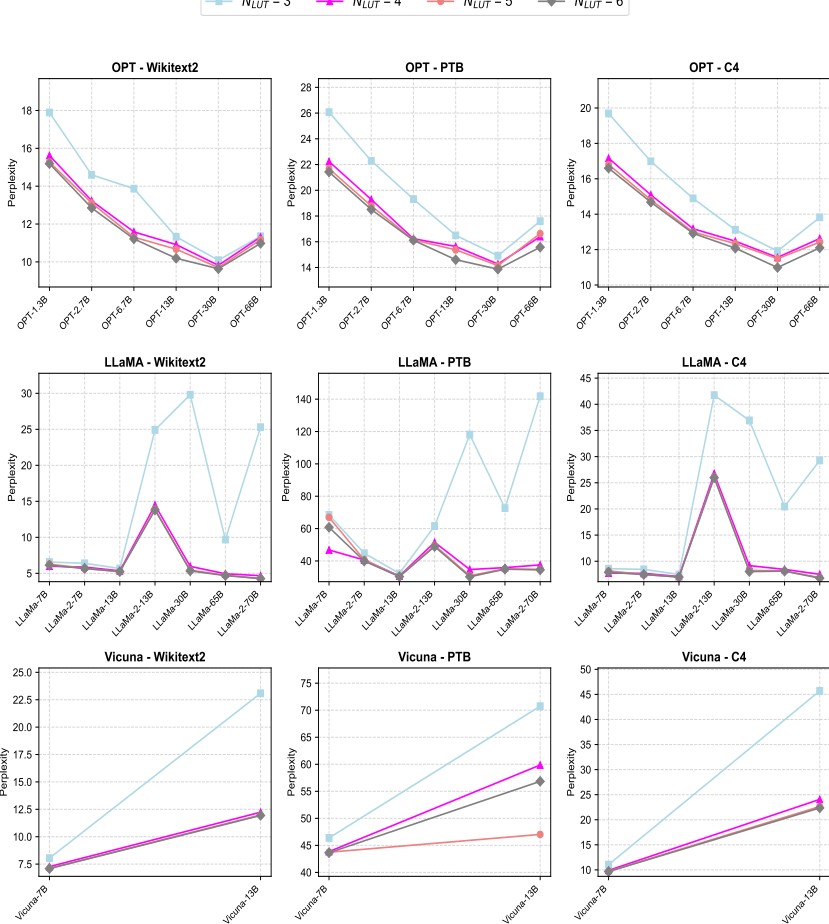

Figure 12: Perplexity impact of varying lookup table (LUT) sizes in SAGE-PTQ across OPT (1.3B–66B), LLaMA-1/2 (7B–70B), and Vicuna (7B, 13B) on WikiText2, PTB, and C4. While OPT performs well with 3-bit LUTs, LLaMA and Vicuna benefit from 4+ bits. A 4-bit LUT offers an effective trade-off between accuracy and storage.

