# OpenReview forum: "Scaling Overhead Matters: Saliency-Aware Graph-Based Efficient Post-Training Quantization for LLMs"
_ICLR.cc/2026/Conference — ICLR 2026 Conference Withdrawn Submission_

### Official Review · Reviewer_9W2K · 2025-10-23

**Soundness:** 2
**Presentation:** 2
**Contribution:** 2
**Rating:** 2
**Confidence:** 4

**Summary:**

In this work, a new post-training quantization algorithm called SAGE-PTQ has been proposed.

After partitioning weights into salient and non-salient weights, the authors further partitioned non-salient weights into multiple clusters, each of which is assigned a separate scale.

To do so, several modules have been proposed; 1) module to judge saliency, 2) graph-based module used for the partitioning of non-salient weights, 3) dual quantizers, etc.

**Strengths:**

The method is clearly explained, but there exists some doubt on practicality of the method.

**Weaknesses:**

1. I have a doubt about the inference speed. After the quantization, each weight matrix of one layer consists of mixed bits where the positions are random. Specifically, in contrast to the standard per-channel or group-wise quantization, weights having the same bit (i.e., salient or non-salient weights) are positioned in random. Also, for non-salient weights where one-bit is assigned, weights sharing the quantization scale are located in random. How to accelerate the inference of such irregularly quantized weights on the real device? Moreover, while the authors compared the inference latency of the original FP16 model and the quantized model obtained via SAGE-PTQ, there is no latency comparison with the standard uniform quantization.

2. An additional concern is that the authors have considered only weights for the partitioning and quantization. However, in almost all PTQ methods, the degradation in the final task loss has been considered much more important than the perturbation in the weight.

3. Comparison with GPTQ is totally unfair. GPTQ assigns a uniform bit to all weights. Also, it seems that the quantization performed in SAGE-PTQ is non-uniform quantization, not the standard uniform quantization. Why did the authors compare their method with GPTQ?

**Questions:**

See Weaknesses

---

> ### Author Response · Authors · 2025-11-21
> **Response Regarding Random Weight Assignment, Inference Speed and Latency Comparison**
>
> We thank the Reviewer for the comments.
>
>
> ***Clarification on “Random” weight Assignment (Q1):***
>
> - Our SAGE-PTQ method does not assign weights randomly. Every single step in SAGE-PTQ method is deterministic.
> - SAGE-PTQ uses two types of groups: one salient group (Module 1, section 3.1) and $N_{\text{uns}}$ inlier groups (Module 2: section 3.2). Each salient weight receives a multi-bit value. Each inlier weight receives a 1-bit value.
> - Different weight groups are represented by indices, where each weight within a group is assigned a group index. Module 5 (section 3.5) applies a fast and efficient approach to encode the indices that reflect weight group assignments in a non-random and efficient way.
> - To clarify how the method is “non-random” and how inference is made fast, module 5 builds a lookup table $L_{\text{LUT}}$​ for each quantized weight matrix.
> - The table $L_{\text{LUT}}$​ is “bit-packed”; the term “bit-packed” means that table elements are compressed to only an average bit-width of $N_{\text{LUT}}$ rather than INT or FP full precision. The table $L_{\text{LUT}}$​ can encode up to $2^{N_{\text{LUT}}}​$ weight groups.
> - Our lookup table $L_{\text{LUT}}$​ is four times smaller in size than regular full-precision lookup schemes since our lookup design only set $N_{\text{LUT}}$ = 4 bits.
> - The elements of  $L_{\text{LUT}}​$ table represent indices in range $[0, 1,\dots$, $2^{N_{\text{LUT}}}-1$]. Each index value encodes one weight group.
>  - During inference, the lookup table is accessed only once from the CPU to recover the group assignment for each weight. This makes the weight group restoration organized, fast, and easy to use during inference.
>
>
> ***Inference Speed and Why SAGE-PTQ Avoids Latency Issues(Q1)***
>
> - SAGE-PTQ does not introduce the inference latency issue. In fact, SAGE-PTQ reduces latency (see section 4.2, end-to-end inference latency experiment).
> - Previous sub-2-bit PTQ methods (BiLLM, PB-LLM) did not analyze inference latency. Past methods split each weight matrix into many position-based blocks (e.g., 128-column blocks). Each block needs its own scale, and inference must repeatedly load and apply these scales. This increases inference latency overhead.
> - In contrast, SAGE-PTQ uses only one row-wise (channel-wise) scale for the salient group and $N_{\text{uns}}​$ scalar scales for all salient (inlier) groups. Module 5 projects these scales into group indices stored in $L_{\text{LUT}}$.
> - During inference, the GPU multiplies the quantized weight matrix $B_l$ with the projected scales using a standard GEMM GPU kernel. This is efficient and fully parallel in GPU device (Table 3).
> - We measure both CPU→GPU lookup transfer time (I/O latency) and the dequantization cost before inference (Table 3). Full real-device latency analysis is given in Section 4.2.
>
>
> ***Latency Comparison With “Standard Uniform Quantization”(Q1)***
>
> - To the best of our knowledge, the term “standard uniform quantization” is not used to describe any established PTQ method for LLMs.
> - Existing sub-2-bit PTQ methods, such as BiLLM and PB-LLM, also do not provide inference-time implementations or latency measurements, so no fair latency comparisons available from prior work. GPTQ reports inference latency, but its GPU kernel is optimized only for the A100 GPU family.
> - In contrast, our SAGE-PTQ lookup-based implementation is device-agnostic and works across GPU types. Our end-to-end latency experiment (section 4.2)  focuses on showing the benefit of reducing model size on device memory, which leads to faster inference and lower hardware requirements.
> - If the reviewer can provide “uniform quantization” baseline with an open-source inference kernel, we will include it in our comparisons.

---

> ### Author Response · Authors · 2025-11-21
> **Clarification on using Weight magnitude metric and GPTQ Comparison**
>
> ***Using Weight Magnitude Instead of Loss-Based Sensitivity (Q2):***
>
> - PTQ methods in the literature typically use one of two metrics to decide saliency and precision assignment: (1) Magnitude-based metrics (e.g., PB-LLM). (2) Sensitivity (loss-based) metrics using Hessian information (e.g., BiLLM, GPTQ).
> - We tested both metrics, magnitude and loss, in SAGE-PTQ. As stated in Module 1 (Section 3.1) and comparison results shown in Appendix C.2 (Figure 11). We Experimentally proved that magnitude-based saliency gave significantly better performance than loss-based saliency.
> - Loss-based methods work for previous methods because they rely on expensive position-based scaling, where each 128-column block receives its own row-wise scale column. This increases untracked scaling overhead.
> - In SAGE-PTQ, we use only 0.004–0.009 average bits for scaling per matrix (section 4.2). To maintain this low scaling overhead, loss-based metrics do not capture the necessary variation in weight magnitude and lead to worse quantization performance. Loss-based metrics also require calibration data, which slows down quantization.
> - Our modular approach achieves strong performance using only weight magnitude saliency metric, without needing calibration or heavy scaling structures.
>
> ***Addressing the Concern About GPTQ Comparison (Q3)***
>
> - To the best of our knowledge, GPTQ does not use or define the term “uniform quantization.” GPTQ assigns the same precision to all weights and applies an asymmetric min–max mapping guided by a Hessian-based loss approximation.
> - We compare SAGE-PTQ to GPTQ for the following reasons.
>
>   1. GPTQ is a weight-only PTQ method that does not require activation quantization, which matches the setting of SAGE-PTQ.
>   2. GPTQ provides low-precision quantization down to 2 bits, while SAGE-PTQ reaches near-binary quantization.
>   3. GPTQ performs layer-wise dequantization on-the-fly during inference. SAGE-PTQ performs band-wise dequantization, where each band is a group of layers.
>   4. GPTQ introduces scaling factors for quantized weights, and SAGE-PTQ also uses scaling factors.
>
> - Following prior sub-2-bit work (PB-LLM, BiLLM), we compare GPTQ and SAGE-PTQ by average weight bit-width and average scaling bit-width. In both comparisons, SAGE-PTQ achieves better WikiText2 perplexity at lower bit-widths (Tables 1 and 2, lines 356–361).

---

> ### Author Response · Authors · 2025-11-25
> **Request to review the rebuttal**
>
> Dear Reviewer 9W2K,
>
> Thank you for taking the time to review our paper. We have addressed your concerns in our submitted response. We kindly request you to review our rebuttal and share any additional comments or concerns you may have. Thank you once again for your valuable feedback!
>
> Authors of submission 22923

---

### Official Review · Reviewer_s8v3 · 2025-10-27

**Soundness:** 3
**Presentation:** 3
**Contribution:** 3
**Rating:** 6
**Confidence:** 4

**Summary:**

The paper introduces SAGE-PTQ, a novel post-training quantization (PTQ) framework designed to compress Large Language Models (LLMs) to near-binary precision while minimizing scaling overhead. The authors identify critical limitations in existing binarization methods, such as reliance on heuristics, rigid weight grouping, and significant untracked memory overhead from scaling factors. The authors conduct extensive experiments on various LLM families (LLaMA, OPT, Vicuna, etc.) and sizes (up to 70B). The results are exceptionally strong, demonstrating significant improvements over state-of-the-art methods like BiLLM and PB-LLM. For instance, on LLaMA-2-7B, SAGE-PTQ achieves a perplexity of 5.87 compared to 32.48 for BiLLM, while reducing GPU memory usage by over 50% and achieving a 1.5x decoding speedup on a 70B model.

**Strengths:**

1. The core contribution—using graph-based clustering on subsampled weights to adaptively determine the number of inlier groups—is highly novel and technically sound. It provides a principled, data-driven alternative to the rigid, block-based grouping schemes prevalent in prior work. This approach is well-motivated by the paper's own statistical analysis of weight distributions across different models and layers (Appendix B).

2. The paper's central theme, that "scaling overhead matters," is timely and crucial. Many works in low-bit quantization focus solely on weight bit-width, ignoring the substantial memory and computational cost of scaling factors. SAGE-PTQ's dual-mode quantizer design, which assigns only one scalar per inlier group, directly and effectively tackles this problem. The reported scaling overhead of ~0.004 bits per weight is a massive improvement over the ~1.0 bit for BiLLM.

3. The authors demonstrate a clear focus on practical deployment. The inclusion of an efficient inference runtime (Module 5), the analysis of memory footprint (Figure 4), and the end-to-end latency case study on a 70B model (Table 3) are commendable. These results show that the benefits are not just theoretical but translate into tangible reductions in memory and latency in resource-constrained scenarios.

4. The paper is well-written, clearly structured, and easy to follow. The modular design is explained logically, with each component's role justified. The ablation studies (Figure 5) effectively isolate and validate the contributions of the novel components (graph clustering and adaptive saliency). The appendices provide valuable supplementary material, including detailed algorithms and the motivating statistical analysis.

**Weaknesses:**

1. The graph-based clustering (Module 2) introduces several hyperparameters, notably the subsampling size and the number of neighbors K in the KNN graph. The paper does not provide an analysis of the method's sensitivity to these choices. The robustness of the "optimal" number of clusters found by the Silhouette Score to these settings is an important, unaddressed point.

2. The processed models in this paper only lies on OPT and LLaMA series, the recent new models such as Qwen, Gemma are missed.

3. SAGE-PTQ is a complex, multi-stage pipeline involving statistical analysis, graph construction, spectral clustering, numerical optimization (Brent's method), and an alternating optimization scheme. While the results justify this complexity, it may pose a barrier to adoption and reproduction. The paper claims faster quantization times than BiLLM (Table 4), which is a strong counter-argument, but the implementation complexity itself remains a concern.

**Questions:**

1. The authors should also discuss the recent SOTA salience-aware papers in related works, such as SliM-LLM[1].

2. I suggest the author refine the design of Figure 1, which is with most blank space, and it is hard to get the details information from this figure.



[1] SliM-LLM: Salience-Driven Mixed-Precision Quantization for Large Language Models. ICML 2025

---

> ### Author Response · Authors · 2025-11-21
> **Response Regarding Graph Hyperparameters, Method Extension, Implementation Complexity and Related Work**
>
> We thank the reviewer for the insightful comments and observations. We will revise the figure layout and add additional comparison results in the updated version of the paper.
>
>
> ***Sensitivity Analysis of Graph-Based Clustering Hyperparameters***
>
> We use a grid search approach to set the hyperparameters for Module 2: graph-based weight grouping.
> - After filtering outliers by module 1, we compute numerical statistics of the salient (inlier) weight distribution and draw a subsample $S_l$​ that reflects these statistics. We test subsampling ratios of [0.01%, 0.03%, 1%, 3%]. A ratio of 0.01% is too small and does not capture the weight distribution for various model sizes. Ratios of 0.03% - 3% give almost identical results. Because larger samples increase runtime, we select $S_l = 0.03 $%$ \times |\Omega_l^{\text{uns}}|$.
> - For the KNN graph, we run a grid search over K $\in$ [1, 20]. For OPT models, stable performance begins at K = 10. For LLaMA, DeepSeek, and Vicuna, stable performance begins at K = 5. This difference is expected because OPT has higher inlier variance than other recent model families, therefore, larger K is needed to reflect weight relations (see Appendix B for weight distribution behavior).
> - We analyze the impact of tuning these hyperparameters on the Silhouette Score and observe only small changes in the optimal number of inlier groups (±1 group). This shows that our group-count selection is robust within the tested ranges.
>
> ***Extending the SAGE-PTQ method to Additional Model Families:***
> - We thank the reviewer for the suggestion. Our experiments focus on OPT and LLaMA because these families cover a wide range of architectures and sizes, and they exhibit strong variation in weight statistics. This allows us to test the robustness of SAGE-PTQ under different distribution patterns.
> - Our method is modular, parametric, and model-agnostic. The saliency filtering, inlier grouping, and quantization modules do not rely on architecture-specific assumptions. We therefore expect SAGE-PTQ to apply to newer families such as Qwen and Gemma.
> - Extending our evaluation to more recent models is a valuable direction for future work.
>
> ***Concerns regarding Implementation Complexity and Reproduction***
>
> - We acknowledge that SAGE-PTQ is a multi-stage pipeline. To keep it efficient, we use a modular and fully parallel implementation. All modules run on the GPU device. Weight matrices across layers are quantized in parallel.
> - Algorithm (1) of Module 2 runs in parallel by evaluating multiple candidate inlier group counts at the same time, which speeds up the selection of the optimal number of groups.
> - Module 3’s quantizer (Mode 2) runs in parallel to binarize different inlier groups.
> For Module 4 saliency thresholding, we use the Brent method since it  uses golden-section search and parabolic interpolation  for faster convergence.
> - As a result, SAGE-PTQ achieves faster quantization than simpler methods (e.g. BiLLM) on the same hardware (Table 4).
>
> ***Discussion of Saliency-Aware Related Work***
>
> - We thank the reviewer for this helpful suggestion. We have reviewed the SliM-LLM approach, which performs saliency-driven mixed-precision PTQ for LLMs. SliM-LLM uses a Hessian-based saliency metric and applies position-based, per-block saliency selection followed by a second-level calibration step to refine outlier detection within each block.
> - However, SliM-LLM applies min–max scaling to every block for both salient and non-salient weights. This introduces significant scaling overhead, averaging about 0.5 bits per weight (similar to PB-LLM, as noted in lines 358–360 of our paper).
> - In contrast, SAGE-PTQ uses far lower scaling overhead while achieving lower WikiText-2 perplexity on most LLaMA model families, compared to SliM-LLM at block size 128 and 2-bit average weight precision, as shown in our results below:
>
> | Method                                   | LLaMa 1-7B | LLaMa 2-7B | LLaMa 1-13B | LLaMa 2-13B | LLaMa 1-30B | LLaMa 1-65B | LLaMa 2-70B |
> |------------------------------------------|------------|------------|-------------|-------------|-------------|-------------|-------------|
> | Slim-LLM (2 bits)                        | 14.58      | 16.01      | 8.87        | 9.41        | 7.33        | 5.90        | 6.28        |
> | **SAGE-PTQ (1.03 bits)** | **5.99**   | **5.87**   | **5.35**    | **14.49**   | **5.97**    | **4.93**    | **7.51**    |
>
> - We will update the related work section to include a discussion of SliM-LLM and clarify these differences.

---

> ### Author Response · Authors · 2025-11-25
> **Request to review the rebuttal**
>
> Dear Reviewer s8v3,
>
> Thank you for taking the time to review our paper. We have addressed your concerns in our submitted response. We kindly request you to review our rebuttal and share any additional comments or concerns you may have. Thank you once again for your valuable feedback!
>
> Authors of submission 22923

---

### Official Review · Reviewer_odiy · 2025-10-31

**Soundness:** 3
**Presentation:** 3
**Contribution:** 2
**Rating:** 6
**Confidence:** 4

**Summary:**

This paper proposes an efficient post-training quantization (PTQ) method, SAGE-PTQ, based on saliency awareness and graph clustering to address memory and latency bottlenecks in the deployment of large language models (LLMs). This method solves the problems of untracked scaling overhead, poor model adaptability, and rigid weight grouping in existing binary PTQ methods through five core modules. Experimental verification shows that SAGE-PTQ performs well on multiple model families (1.3B-70B parameters) such as LLaMA, OPT, and Vicuna, with an average weight bit width of only 1.03 bits and a low scaling overhead of 0.004 bits. Compared with existing methods such as BiLLM, SAGE-PTQ reduces confusion by more than 77%, GPU memory usage by 50%, and inference speed by 1.5 times, providing an efficient solution for deploying LLMs on edge devices.

**Strengths:**

1. The motivation is clear and meaningful, focusing on the core points of LLM binary PTQ.

2. The proposed methods are highly compatible with the problem-solving objectives.

3. The experimental design is comprehensive and shows relatively good performance, and with calibration costs and inference speed reports.

**Weaknesses:**

1. The saliency measurement based on Hessian has been thoroughly validated by works such as BiLLM and GPTQ, and the conclusion in the paper shows that the magnitude-based method is significantly better. What is the reason of this phenomenon and is it related to binary quantization settings?

2. Why was sparse KNN chosen for clustering instead of other methods, and more quantitative experiments or metrics should be explained.

3. Dual mode quantization introduces complex scaling factors. Will this incur additional overhead for actual inference, or will it adapt to existing inference frameworks or require operator redesign?

4. The paper should provide an individual ablation study for different components.

**Questions:**

See weaknesses

---

> ### Author Response · Authors · 2025-11-21
> **Response Regarding Magnitude Saliency, Sparse KNN Graph Motivation, Quantization overhead and Ablation**
>
> We thank the reviewer for the helpful comments.
>
>
> ***Reasoning Behind Magnitude-based Saliency Metric (Q1):***
> - We evaluated Module (1) of SAGE-PTQ using both magnitude-based saliency and Hessian-based saliency, as stated in Section 3.1 (lines 164–167). Our experiments show that magnitude-based saliency yields much better perplexity after quantization (Figure 11, Appendix C.2).
> - We believe the Hessian-based approach works only under certain assumptions. Prior results can be explained by how previous methods use expensive scaling. Methods such as BiLLM and GPTQ rely on position-based scaling, assigning a separate channel-wise scale to every block of 128 columns. This heavy scaling lets Hessian-based saliency correct local errors, but it also adds large, untracked scaling overhead
> - SAGE-PTQ uses a simpler and much smaller scaling scheme. We apply one row-wise scale for all salient weights. We use only $N_{\text{uns}}$​ scalar scales for all unsalient groups. This results in only 0.004–0.009 average bits of scaling per matrix. Under this low-overhead design, Hessian-based saliency becomes ineffective since it cannot capture the true tail behavior of the weight distribution. Hessian-based saliency also cannot isolate the weights that create long-tail outliers, which is essential for binary quantization.
> - Since Magnitude-based saliency aligns naturally with the weight statistical structure, Magnitude metric produces better results under our quantization setting.
>
>
> ***Justification for the Sparse KNN Graph Approach***
> - Module 2 uses a sparse KNN graph to determine the optimal number of unsalient (inlier) weight groups. We choose a graph-based method because it offers advantages not found in similar approaches (such as K-means), for the following reasons:
>   1. preserves affine relationships among inlier weights so that each group contains weights with similar magnitudes.
>   2. Allows us to use the silhouette score as a stable clustering-quality metric for selecting the optimal group count.
>   3. Enables efficient construction of an affinity matrix using a sparse KNN graph over a subsampled set of inlier weights.
> - We also evaluated K-means as an alternative. However, K-means is sensitive to initialization, especially with the large number of inlier weights in LLMs. K-means produced inconsistent clustering across runs and poorer clustering quality. K-means also performed poorly on subsampled weights and collapsed most weights into a few clusters, making silhouette scores unreliable. Applying K-means to the full inlier set would be computationally expensive.
> - For the above mentioned reasons and because of K-means limitations, sparse KNN graph clustering is a more reliable and efficient method for determining the optimal number of inlier groups.
>
>
> ***Scaling Overhead and Compatibility of Dual-Mode Quantization***
> - SAGE-PTQ fully accounts for scaling and lookup overhead during quantization. Dual-mode quantization does not require operator redesign. It adapts to existing inference frameworks.
> - Mode 1 of Module 3 (section 3.3, line 230) assigns one channel-wise (row-wise) scale for all salient weights. This adds only a single scale column per matrix.
> - Mode 2 of Module 3 (section 3.3, line 250) assigns one scalar scale per inlier group, resulting in $N_{uns}$ total scalar scales for the entire unsalient set.
> - Because of module 3 design, the total scaling overhead remains extremely small, only 0.004–0.009 bits per weight across model families. All scale redistribution and lookup logic is handled by Module 5 during inference (see Section 3.5 for module 5 logic and 4.2 for inference results).
> - The method therefore introduces minimal overhead and integrates cleanly with standard GPU inference pipelines.
>
>
> ***Clarification on Ablation Study:***
>
> We provided a comprehensive ablation analysis in Section 4.3 of the paper.
> - We evaluate the individual and joint effects of graph-based grouping and adaptive saliency thresholding (Figure 5). We show that both modules are necessary for strong performance.
> - We also study different lookup sizes and find that $N_{LUT} = 4$ is the minimum suitable choice, since performance remains stable for larger $N_{LUT}$ values across model families (Figure 6).
> - Additional ablation results regarding saliency metric selection and lookup size impact are included in Appendix C.2 and Appendix D, respectively.

---

> ### Author Response · Authors · 2025-11-25
> **Request to review the rebuttal**
>
> Dear Reviewer odiy,
>
> Thank you for taking the time to review our paper. We have addressed your concerns in our submitted response. We kindly request you to review our rebuttal and share any additional comments or concerns you may have. Thank you once again for your valuable feedback!
>
> Authors of submission 22923

---

### Official Review · Reviewer_i5HS · 2025-11-01

**Soundness:** 3
**Presentation:** 3
**Contribution:** 3
**Rating:** 4
**Confidence:** 5

**Summary:**

The paper presents SAGE-PTQ, a post-training quantization method for large language models that pushes weight precision down to nearly 1-bit while maintaining strong performance. The method includes several components: saliency-aware outlier detection, graph-based clustering for the remaining weights, a dual-mode quantizer, adaptive outlier thresholding, and a bit-packed inference kernel.

**Strengths:**

This is a well-engineered solution to a real problem. The design is thoughtful, the evaluation is thorough, and the empirical improvements over prior 1-bit methods like BiLLM are impressive. The paper is particularly strong in showing practical deployment benefits, like reduced memory and faster decoding, and the implementation seems polished.

**Weaknesses:**

The novelty feels somewhat incremental. The overall idea, ie, handling outliers in higher precision and binarizing the rest, has been done before, and the contributions here read more like refined heuristics than a conceptual breakthrough. The graph-based grouping and adaptive thresholding are reasonable, but not clearly game-changing. The exposition is dense and could be clearer in places, especially around the clustering and threshold modules. It’s also a bit disappointing that edge deployment is heavily emphasized, yet all experiments are on GPUs. Without tests on CPUs or mobile accelerators, the “edge-ready” claim feels premature.

A few other gaps: there’s little discussion of quantization time or graph construction cost, which could be substantial for large models. Some baseline comparisons are missing, especially to popular 3–4 bit methods like GPTQ or AWQ, which would help contextualize how close this 1-bit method gets to higher-bit PTQ in accuracy. Also, the method involves many components and hyperparameters, but the paper says little about sensitivity or robustness.

Overall, this is a strong paper in terms of engineering and results, but the conceptual advances feel modest. I lean weak reject for now, mostly due to the limited novelty and unclear practical trade-offs during quantization. With clearer exposition, a deeper novelty discussion, and broader baselines, I could see this moving toward acceptance.

**Questions:**

1. How exactly is the saliency threshold determined? Do you optimize per layer, and how expensive is that?
2. Did you compare graph grouping to simpler clustering like K-means?
3. How sensitive is performance to the outlier bit budget or saliency percentage?
4. Any insights into how well the bit-packing and lookup work on CPUs or other edge devices?
5. How does this compare in accuracy to 3–4 bit quantization methods?

---

> ### Author Response · Authors · 2025-11-21
> **Clarification Regarding Novelty Concerns**
>
> We thank the reviewer for the detailed comments.
>
> We agree with the reviewer that handling outliers in higher precision and binarizing the rest is standard idea. However, SAGE-PTQ modules address specific unresolved problems in post-training quantization of LLMs. Our method is not a heuristic refinement. Every design choice is motivated by theoretical reasoning and validated through experiments \
> \
> **Module 1: Saliency-guided filtering (section 3.1):**
> - **Problem addressed**: (a) identify salient (outlier) weights in a model-agnostic way. (b) respect the statistical distribution of each weight matrix (lines 168–172). (c ) detect outliers without assuming  fixed percentage of important weights across layers
> - **Prior work limitations:** rely on loss-based Hessian metrics that require calibration data (GPTQ, BiLLM). Other methods use a fixed percentage of high-magnitude weights across all layers (PB-LLM). These approaches ignore the large variation in weight statistics across model families and matrices.
> - **Our Approach:** module 1 (a) models weights distribution of each matrix. (b) isolates values causing long-tail behavior of the histogram. **Novelty**: use statistical structure of each matrix to detect outliers in a data-free and distribution-aware manner.
>
> **Module 2: Sparse graph modeling for affinity-based weight grouping (section 3.2):**
> - **Problem addressed:** determining how many unsalient (inlier) weight groups are needed so each group contains weights with similar magnitudes.
> - **Prior work limitations:** (GPTQ, PB-LLM, BiLLM) use position-based blocks that ignore the true weight distribution and introduce redundant scaling.
> - **Our approach:** (a) models the inlier weights with subsampled representative weights values from each matrix, (b) build a KNN graph out of representative weights,(c ) run spectral clustering on KNN graph with several candidate group counts, (d) select the number that maximizes a clustering quality metric (silhouette score, Eq. (2)). (e)  group overall inlier weights using the optimized group based on their statistical quantiles. **Novelty**: the number of inlier groups is adaptive, data-driven, and optimized per matrix rather than fixed. We use graph modeling rather than simpler methods (e.g., K-means) because we proved the graph approach is robust to initialization and captures attention patterns among weights (see detailed answer below: Comparison of Graph Grouping to Similar Clustering Methods). Detailed pipeline of module 2 is shown in Algorithm 1, Appendix A.
>
> **Module 3: Optimized Dual-model weight Quantization (section 3.3):**
> - **Problem addressed:** assigning precision to salient and inlier weights in a way that minimizes approximation error and scaling overhead.
> - **Prior work limitations:** use min–max scaling for multibit weights (GPTQ, BiLLM, PB-LLM) and sign-based binarization for unsalient weights, which is suboptimal in terms of quantization error and introduces large, untracked scaling overhead.
> - **Our approach:** (a) formulate weight quantization as a discrete optimization problem. (b) For salient weights, solve Eq. (6) under the constraint that the salient group uses a single channel-wise scale. (c ) For inlier weights, solve Eq. (8) under the constraints that each inlier group is binarized and assigned a single scalar scale. **Novelty:** proposed novel algorithm to solve quantization optimization problem. We apply a relaxation method and design an iterative algorithm that finds scale values with minimal overhead. (See algorithm 2, appendix A).
>
> **Module 4:  Saliency allocation via adaptive thresholding (section 3.4) (Q1):**
> - **Problem addressed:** adaptively selecting percentage of salient weights while accounting for variation across model families and matrices.
> - **Prior work limitations:** split weights into salient and unsalient sets using fixed percentages (PB-LLM) or heuristic searches (BiLLM). These approaches are suboptimal in performance and can misidentify outliers.
> - **Our approach:** (a) formulates saliency thresholding as a numerical optimization problem solved with efficient Brent method. (b) searches within $[0, p^{\text{sal}}_{\max}]$ (c ) selects the percentage of salient weights that minimizes the relative error between quantized and original weight matrix under module 3 quantization scheme. **Novelty**: saliency thresholding is adaptive and optimized per matrix, and does not rely on fixed heuristics.
>
> **Module 5: Efficient Inference Runtime (section 3.5):**
> - **Problem addressed:** efficiently assigning scales to weight groups during inference, especially when computation is split between the CPU and the GPU.
> - **Prior work limitations:** (BiLLM, PB-LLM) do not address lookup efficiency or deployment constraints.
> - **Our approach:** builds a lookup table that enables fast scale retrieval during inference. **Novelty:** a deployment-ready lookup mechanism that supports efficient CPU–GPU interaction.

---

> ### Author Response · Authors · 2025-11-21
> **Clarification on Quantization Runtime, Graph Construction Cost and Comparison to Similar Clustering Approaches**
>
> ***Discussion on Quantization Runtime:***
> - We acknowledge that SAGE-PTQ has a complex pipeline. To make it efficient, we use a modular and parallel implementation. All modules (explained above) are compatible to run fully on the GPU device. Weight matrices across layers are quantized in parallel.
> - Algorithm (1) of Module (2) runs in parallel (see appendix A) by testing multiple candidate unsalient (inlier) group counts at the same time. This speeds up the selection of the optimal number of groups.
> - Module 3’s quantizer (Mode 2, line 250) binarizes unsalient (inlier) groups in parallel.
> - For Module 4 saliency thresholding, we use Brent method, which uses golden-section search and parabolic interpolation for faster convergence than other options.
> - As a result, our method runs faster than simpler approaches such as BiLLM on the same hardware, as shown in table 4 in the paper
>
>
> ***Discussion on Graph Construction Cost:***
>
> - We achieve efficient construction of Sparse KNN graphs because we use subsampled data. Our construction and hyperparameters preserve weight structure and ensure fast implementation. We use a grid search approach to set the hyperparameters for Module 2.
> - After filtering outliers by module 1, we compute numerical statistics of the unsalient (inlier) weight distribution and draw a subsample $S_l$​ that reflects these statistics.
> - We test subsampling ratios of [ 0.01 %, 0.03 %, 1 %, 3 %] . A ratio of 0.01% is too small and does not capture the weight distribution for various model sizes. Ratios of 0.03% - 3% give almost identical results. Because larger samples increase runtime, we select $S_l =$ 0.03 % $\times |\Omega_l^{\text{uns}}|$.
> - For the KNN graph, we run a grid search over $K \in [1, 20]$. For OPT models, stable performance begins at K = 10. For LLaMA, DeepSeek, and Vicuna, stable performance begins at K = 5. This difference is expected because OPT has higher inlier variance than other recent model families, therefore, larger K is needed to reflect weight relations (see Appendix A).
>
> ***Comparison of Graph Grouping to Similar Clustering Approaches (K-means). (Q2)***
> - We also tested K-means for Module 2. Recall that module 2 adaptively determines the number of inlier groups unique to every matrix. We replaced the graph-based clustering step with K-means while keeping the rest of module (2) the same.
> - K-means exhibited clear limitations. It is sensitive to initialization, and this becomes worse when the number of inlier weights is large, as in LLMs. We saw inconsistent perplexity results across runs and lower clustering quality. On subsampled weights, K-means often collapsed most points into a few clusters and left other clusters nearly empty.
> - K-means limitations produced unreliable silhouette scores (clustering quality metric, Eq(2)). Therefore, we could not use K-means to select the optimal number of salient (inlier) groups.
> - Our graph-based method (section 3.2) avoids the mentioned K-means problems. KNN graphs are efficient to compute on subsampled weights. Spectral clustering is stable, less sensitive to initialization, and captures local weight structure better than K-means. This makes the graph-based method more reliable and efficient for determining the number of inlier groups.

---

> ### Author Response · Authors · 2025-11-21
> **Discussion on Method Sensitivity, Deployment on CPU and Comparison to Higher-precision Baselines**
>
> ***Method Robustness and Sensitivity to Hyperparameters (Q3):***
>
> We analyzed method sensitivity to various hyperparameter selections.
>
> - **Maximum Saliency Percentage:**  we analyze weight distributions of several model families (Appendix B) to determine maximum saliency percentage. We test percentages [0.01, 0.05, 0.1, 0.5]. Module 4 consistently selects values below 0.1 for the OPT family and below 0.01 for LLaMA, DeepSeek, and Vicuna. These results align with the model statistics: OPT has shorter tails and larger variance (more outliers of smaller magnitude), while LLaMA, DeepSeek, and Vicuna have longer tails and small variance (fewer but larger outliers). We use KL divergence to quantify distribution variation, as shown in Appendix A (Figures 7–9).
> - **Outlier Bit Budget:** performance improves as the bit budget increases, so we establish design rules to select the minimum precision that preserves accuracy. Based on this analysis, we set 0.01 saliency percentage and a 4-bit outlier budget for long-tail models (LLaMA 7B–70B, DeepSeek 7B, Vicuna 7B/13B) and 0.1 and 2-bit for short-tail models (OPT 1.3B–66B).
> - **Lookup Size:** We test lookup bit budgets $N_{\text{LUT}}​$ in the range [2,3,4,5,6]. Each budget allows up to $2^{N_{\text{LUT}}}​$ weight groups. For all model families, performance is stable when $N_{\text{LUT}} > 2$. Results are unchanged for values above 3. We therefore set $N_{\text{LUT}} = 4$ as a robust default. Additional lookup analysis is provided in Section 4.3 and Appendix D.
>
> The above mentioned hyperparameters settings represent minimal settings, and the method remains robust and improves further with larger budgets.
>
> \
> ***Deployment cost on CPU and other edge devices (Q4):***
> - We appreciate the reviewer’s suggestion and acknowledge that our mention of “edge devices” may have caused confusion. In the final version, we will remove the emphasis on edge deployment to avoid misinterpretation.
> - Our method focuses on reducing the computation and memory demands of large LLMs so they can run on resource-limited GPU setups, not on CPU-only environments.
> - While our lookup strategy is parallelizable and can run on multi-core CPUs, CPU-only inference is not a realistic target for models in the 7B–70B range due to the heavy matrix operations required. To the best of our knowledge, such models are not deployed purely on CPUs.
> - We addressed primary concerns of GPU memory limits and the I/O overhead between CPU and GPU during inference. Our proposed bit-packing and lookup mechanism is designed to minimize this overhead and enable faster layer loading during token decoding, as described in Section 4.2. This makes the method practical for single-GPU, memory-constrained scenarios, which are the realistic “edge” cases for current LLMs.
>
> \
> ***Comparison to Higher-Precision Baselines (Q5)***
>
> - Our comparisons focus on methods targeting sub-2-bit precision (section 4.2). We also evaluated our method against higher-precision baselines and the unquantized FP16 model.
> - SAGE-PTQ provides reasonable linguistic output compared to FP16 with minimal perplexity degradation. It outperforms GPTQ at 2-bit and 3-bit settings with block size 128. We further compare to AWQ at 3 bits and block size 128 on the OPT family using WikiText perplexity. Our method performs better on models below 13B and achieves similar results on larger models while using only 36% of the precision required by AWQ. The results are shown below
> - We will add these results and higher-precision baselines to the updated version of the paper.
>
>
>
> | Method                | OPT 1.3B | OPT 2.7B | OPT 6.7B | OPT 13B | OPT 30B | OPT 66B |
> |-----------------------|----------|----------|----------|----------|----------|----------|
> | FP16                  | 14.62    | 12.47    | 10.86    | 10.13    | 9.56     | 9.34     |
> | GPTQ (3 bits)         | 20.97    | 16.88    | 14.86    | 11.61    | 10.27    | 14.16    |
> | AWQ (3 bits)          | 16.32    | 13.58    | 11.39    | 10.56    | 9.77     | 9.62     |
> | **SAGE-PTQ (1.07 bits)** | **15.63** | **13.23** | **11.59** | **10.91** | **9.82** | **11.31** |

---

> ### Author Response · Authors · 2025-11-25
> **Request to review the rebuttal**
>
> Dear Reviewer i5HS,
>
> Thank you for taking the time to review our paper. We have addressed your concerns in our submitted response. We kindly request you to review our rebuttal and share any additional comments or concerns you may have. Thank you once again for your valuable feedback!
>
> Authors of submission 22923

---

### Comment · Area_Chair_Ma4g · 2025-11-24

Dear Reviewers,

**We kindly encourage you to review and respond to the authors’ rebuttals**. Your timely feedback is important for ensuring a fair and thorough review process. Thank you for your contributions to ICLR 2026.

AC

---

### Note · Authors · 2026-05-05

I have read and agree with the venue's withdrawal policy on behalf of myself and my co-authors.

---

### Meta-Review · Area_Chair_JP24 · 2025-12-21

**Summary:**

The paper presents SAGE-PTQ, a post-training quantization framework for large language models that reduces weight precision to nearly 1-bit while maintaining strong performance. The method achieves this feat by leveraging a combination of saliency-aware outlier detection, graph-based clustering of non-salient weights, a dual-mode quantizer, adaptive thresholding, and a bit-packed inference kernel. Extensive experiments across models such as LLaMA, OPT, and Vicuna demonstrate significant improvements in GPU memory usage and decoding speed, thereby providing a promising pathway for edge deployment.

**Reviewer Concerns:**

Despite these engineering strengths and clear empirical benefits, several reviewers raised concerns. The novelty of the approach is perceived as incremental, with the contributions resembling refined heuristics rather than a conceptual breakthrough. Reviewers pointed out that many components, such as saliency measurement and graph clustering, have precedents in prior work, and questions remain about the sensitivity to hyperparameters, inference performance on devices beyond GPUs, and the fairness of comparisons with standard and 3–4 bit quantization methods. Additionally, clarity issues in the exposition of key components and the lack of an in-depth analysis of quantization time and graph construction cost further complicate the assessment.

**Reviewer Scores:**

While the reviewers commend significant reductions in memory footprint and improvements in decoding speed, their evaluations range from marginal acceptance to rejection. Overall, the aggregated sentiment tilts toward a borderline rejection. The work, though promising for practical deployment, requires a more precise exposition of its novel contributions, additional sensitivity analyses, and more comprehensive baseline comparisons to justify acceptance robustly.

---

### Decision · Program_Chairs · 2026-01-26

Reject